# Single-cell proteomics reveals changes in expression during hair-cell development

**Ying Zhu[1†], Mirko Scheibinger[2†], Daniel Christian Ellwanger[2,3], Jocelyn F Krey[4,5], Dongseok Choi[6,7], Ryan T Kelly[1,8], Stefan Heller[2], Peter G Barr-Gillespie[4,5]***

[1]Environmental Molecular Sciences Laboratory, Pacific Northwest National Laboratory, Richland, United States; [2]Department of Otolaryngology Head and Neck Surgery, Stanford University, Stanford, United States; [3]Genome Analysis Unit, Amgen Research, Amgen Inc, South San Francisco, United States; [4]Oregon Hearing Research Center, Oregon Health & Science University, Portland, United States; [5]Vollum Institute, Oregon Health & Science University, Portland, United States; [6]OHSU-PSU School of Public Health, Oregon Health & Science University, Portland, United States; [7]Graduate School of Dentistry, Kyung Hee University, Seoul, Republic of Korea; [8]Department of Chemistry and Biochemistry, Brigham Young University, Provo, United States

**\*For correspondence:**
gillespp@ohsu.edu

[†]These authors contributed equally to this work

**Abstract** Hearing and balance rely on small sensory hair cells that reside in the inner ear. To explore dynamic changes in the abundant proteins present in differentiating hair cells, we used nanoliter-scale shotgun mass spectrometry of single cells, each ~1 picoliter, from utricles of embryonic day 15 chickens. We identified unique constellations of proteins or protein groups from presumptive hair cells and from progenitor cells. The single-cell proteomes enabled the de novo reconstruction of a developmental trajectory using protein expression levels, revealing proteins that greatly increased in expression during differentiation of hair cells (e.g., OCM, CRABP1, GPX2, AK1, GSTO1) and those that decreased during differentiation (e.g., TMSB4X, AGR3). Complementary single-cell transcriptome profiling showed corresponding changes in mRNA during maturation of hair cells. Single-cell proteomics data thus can be mined to reveal features of cellular development that may be missed with transcriptomics.
DOI: https://doi.org/10.7554/eLife.50777.001

## Introduction

Hair cells, the sensory cells of the inner ear, carry out a finely orchestrated construction of an elaborate actin cytoskeleton during differentiation. Progenitors of vestibular hair cells, the supporting cells (*Roberson et al., 1992*), have an unremarkable actin cytoskeleton. By contrast, differentiating hair cells express a wide array of actin-associated proteins, including crosslinkers, membrane-to-actin linkers, and capping molecules, and use them to rapidly assemble mechanically sensitive hair bundles on their apical surfaces (*Shin et al., 2013*; *Ellwanger et al., 2018*). Hair bundles consist of ~100 stereocilia each, filled with filamentous actin (F-actin) and arranged in multiple rows of increasing length; by maturity, stereocilia contain >90% of the F-actin in a hair cell (*Tilney and Tilney, 1988*).

Along the axis of the chicken cochlea, stereocilia systematically decrease in maximum height (from >5 to 1.5 μm) and increase in number per cell (from 30 to 300) as the frequency encoded increases (*Tilney et al., 1992*). Despite these changes, a quantitative analysis suggested that within experimental error, hair cells use the same amount of actin to build these disparate hair bundles (*Tilney and Tilney, 1988*). There is little evidence of substantially increased expression of actin genes

during hair cell differentiation (*Ellwanger et al., 2018*), suggesting that hair cells use the existing monomeric actin that is available during differentiation to build a bundle (*Tilney and Tilney, 1988*).

The number of hair cells in the chick utricle increases 15-fold from embryonic day 7 (E7) to post-hatch day 2, with hair-cell production peaking at E12 (*Goodyear et al., 1999*). Because of this asynchronous development, at any given age, cells are in distinct states along the pathway from progenitors to hair cells. While bulk sampling of cells averages over these developmental distinctions, sampling and analyzing individual cells of the E15 chick utricle allowed for transcriptomics examination of a large portion of the developmental trajectory for forming hair cells from supporting cells (*Ellwanger et al., 2018*), and should allow for a corresponding trajectory analysis using proteomics.

We sought to understand how supporting cells, with their modest F-actin cytoskeletons, could transform rapidly into stereocilia-endowed hair cells without significant upregulation of actin gene transcription. We used a highly sensitive single-cell proteomics approach to assess the concentrations of the abundant proteins in cells isolated from the E15 chick utricle. We found that hair cells were readily distinguished from supporting cells based on only 50–75 proteins identified in each. Notably, the actin monomer binding protein thymosin β4 (TMSB4X) was abundant in supporting cells but not in hair cells, and its expression decreased as progenitors developed into hair cells, which we revealed using trajectory analysis based on the proteomics data. Single-cell RNA-seq (scRNA-seq) analysis showed that *TMSB4X* transcripts are downregulated when transcription of *ATOH1*, a key regulator of hair cell differentiation, is activated. These data are consistent with a model that suggests that existing monomeric actin is made available for hair-bundle assembly by the degradation of TMSB4X.

## Results

### Single-cell proteomics applied to E15 chick utricle hair cells

Recent reports demonstrated that extensive shotgun mass spectrometry characterization of the proteins of single cells is possible when samples are processed in nanoliter-scale volumes (*Zhu et al., 2018b*; *Zhu et al., 2018c*; *Zhu et al., 2018a*; *Zhu et al., 2018d*). About 700 proteins or protein groups could be detected from a single HeLa cell (*Zhu et al., 2018a*), which is estimated to have a volume of a few picoliters (*Zhao et al., 2008*; *Park et al., 2008*). We analyzed single cells from E15 chicken utricle; by using peeled epithelia for cell dissociation, we limited the cell types analyzed to hair cells and supporting cells (*Herget et al., 2013*), each of which is smaller than a HeLa cell. To distinguish the two cell types, we labeled utricle cells with FM1-43, which labels hair cells more strongly than supporting cells (*Herget et al., 2013*; *Ellwanger et al., 2018*). After cell dissociation, we collected single cells and pools of 3, 5, and 20 cells in nanowells using fluorescence-activated cell sorting (FACS; *Figure 1A* and *Figure 1—figure supplement 1*). As expected, collected cells with high levels of FM1-43 (FM1-43$_{high}$) had hair bundles and elongated cell bodies (*Figure 1B*), both of which are characteristic of hair cells; FM1-43$_{low}$ cells—mostly supporting cells—were round after FACS (*Figure 1C*). We measured cell volumes after cell sorting. FM1-43$_{high}$ cells averaged 1.01 ± 0.20 picoliters (mean ± SD; $N = 9$), while FM1-43$_{low}$ cells averaged 0.67 ± 0.15 picoliters ($N = 8$). For comparison, using the same method, we measured the volume of HeLa cells to be 4.9 ± 1.1 picoliters ($N = 5$).

We used the nanoPOTS (nanodroplet processing in one-pot for trace samples) approach to carry out all sample processing steps in single nanowells (*Zhu et al., 2018b*). After protein extraction, reduction, alkylation, and proteolysis, digested peptides were collected and separated using nano-liquid chromatography on a 30-μm-i.d. column (*Zhu et al., 2018b*) (*Figure 1A*). The separated peptides were delivered to an Orbitrap Fusion Lumos Tribrid mass spectrometer and were analyzed using data-dependent acquisition. Peptides were identified, quantified, and assembled into proteins with Andromeda and MaxQuant (*Cox and Mann, 2008*; *Cox et al., 2011*), using Match Between Runs (*Tyanova et al., 2016*; *Zhu et al., 2018b*) to identify unmatched peptides based on their accurate masses and liquid chromatography retention times. To quantify proteins based on molar abundance, we used intensity-based absolute quantification (iBAQ), calculated from the sum of peak intensities of all peptides matching to a specific protein divided by the number of theoretically observable

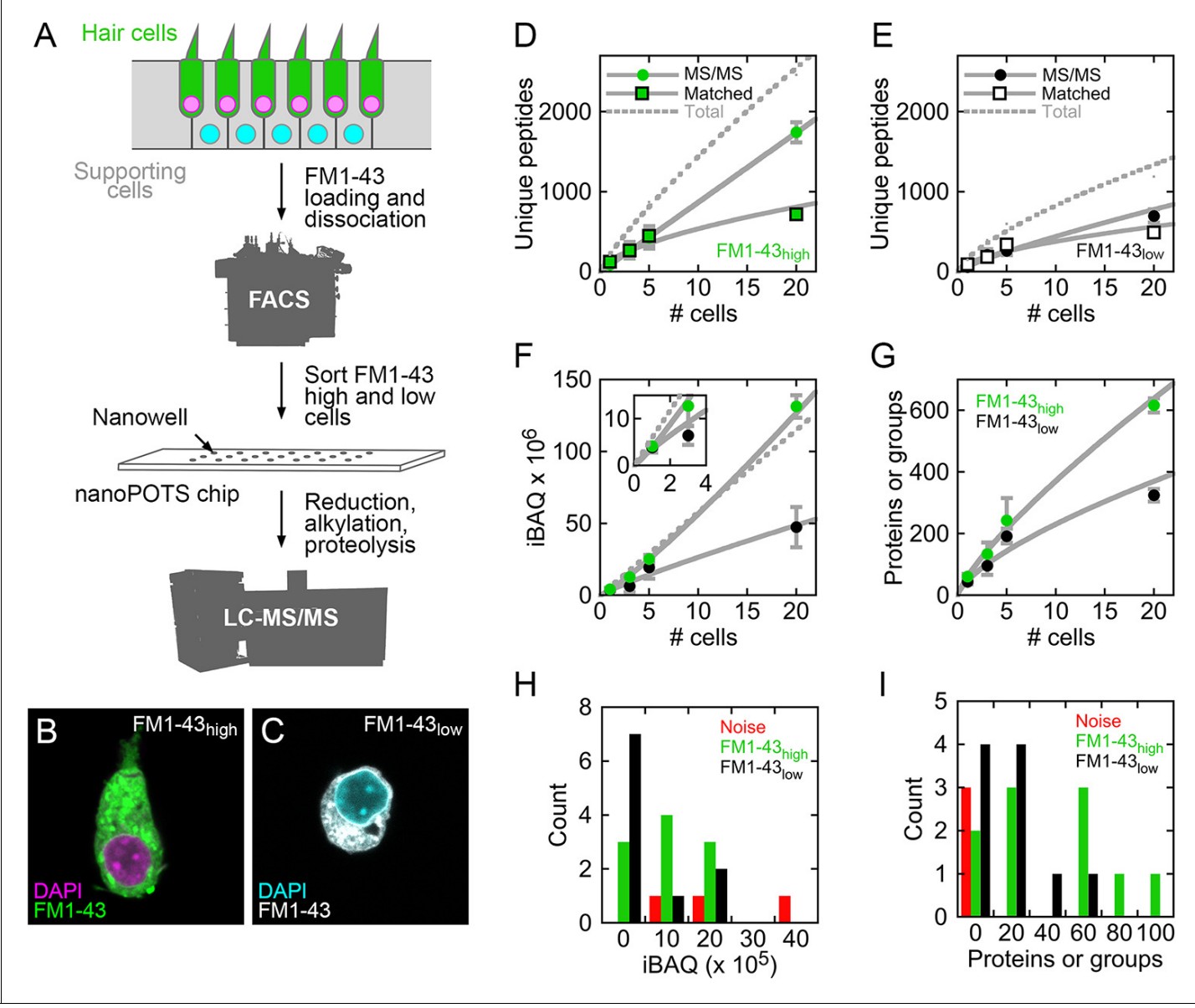

**Figure 1.** Mass spectrometry of single cells and small cell pools from E15 chick utricle. (A) Experimental design. The E15 chick utricle's sensory epithelium consists of sensory hair cells and supporting cells, which are also progenitor cells. FM1-43 labels hair cells more strongly than supporting cells. The dissociated cells were sorted by FACS and deposited into single nanowells in nanoPOTS chips, where sample processing was carried out without transfer. Samples were loaded into glass microcapillaries and were analyzed by mass spectrometry. LC-MS/MS, liquid chromatography-tandem mass spectrometry. (B) FACS-sorted FM1-43$_{high}$ cell with typical hair cell cytomorphology, including apical hair bundle. (C) FACS-sorted FM1-43$_{low}$ cell shows rounded cytomorphology after dissociation. For B and C, FM1-43 dye was added again to cells after sorting in order to visualize cell shape; dye intensity therefore is not representative of the signal used for sorting hair cells from supporting cells. (D–E) Relationship between number of cells and unique peptides. Peptides directly identified by MS2 (peptide fragmentation) spectrum matching are shown by circles and those indirectly identified by Match Between Runs by squares; data are separately plotted for FM1-43$_{high}$ (D) and FM1-43$_{low}$ (E). Gray solid lines are power fits to data through (0,0); gray dashed line is fit to sum of the MS/MS and Matched data. (F) Relationship between number of cells and total iBAQ. Gray solid line is power fit through (0,0); gray dashed line is linear fit through (0,0). Green, FM1-43$_{high}$; black, FM1-43$_{low}$. Inset shows 1–3 cells only. (G) Relationship between number of cells and the total number of proteins or protein groups identified. Gray solid line is power fit through (0,0). Data for D-G were from Experiment 1; mean ± SEM are plotted. (H) Distribution of total iBAQ for individual cells (FM1-43$_{high}$, green; FM1-43$_{low}$, black) or for blank wells (red). Count refers to the number of cells in a bin. Note that total iBAQ does not distinguish individual cells from noise. (I) Distribution of number of identified proteins or protein groups. This measure distinctly distinguishes individual cells from noise; those FM1-43$_{high}$ or FM1-43$_{low}$ samples with low numbers of identifications likely do not have cells in the nanowells. Data for H-I were from Experiment 2.

DOI: https://doi.org/10.7554/eLife.50777.002

*Figure 1 continued on next page*

*Figure 1 continued*

The following source data and figure supplements are available for figure 1:

**Source data 1.** MaxQuant analysis of single cell proteomics data.

DOI: https://doi.org/10.7554/eLife.50777.005

**Figure supplement 1.** Characterization of isolated utricle cells.

DOI: https://doi.org/10.7554/eLife.50777.003

**Figure supplement 2.** Peptide coverage of TMSB4X in mass spectrometry experiments.

DOI: https://doi.org/10.7554/eLife.50777.004

peptides (*Schwanhäusser et al., 2011*). All mass spectrometry data are deposited at ProteomeXchange, and analyzed data are reported in *Figure 1—source data 1*.

About 200 unique peptides were identified in each single FM1-43$_{high}$ cell; about half were identified by MS/MS scans and half by matching (*Figure 1D*). The total number of unique peptides increased to ~2500 in pools of 20 cells, with about 70% identified by MS/MS scans (*Figure 1D*). Only ~1200 peptides were identified in pools of 20 FM1-43$_{low}$ cells, with about 60% identified by MS/MS scans (*Figure 1E*). Total iBAQ rose nonlinearly with the number of cells (*Figure 1F*), suggesting that some protein was lost to surface adsorption; while small relative to typical sample wells, the volume of the nanowell is still 50,000-fold larger than the volume of a utricle cell. Because sample processing occurred in a protected nanowell environment using robotic liquid handling, the total iBAQ attributed to keratins (e.g., human skin contamination) was only ~0.1% of the total, far less than >50% occurring in some mass-spectrometry experiments with small amounts of protein. The number of proteins or protein groups identified increased from ~60 for FM1-43$_{high}$ single cells to nearly 600 for pools of 20 cells (*Figure 1G*); fewer proteins were identified in supporting cells, likely because of their smaller volume.

Comparison of single FM1-43$_{high}$ and FM1-43$_{low}$ cells to wells with collection triggered to noise allowed us to confirm the presence of single cells, even without visual inspection of the wells. Total iBAQ did not accurately indicate which wells contained single cells (*Figure 1H*), presumably because the total signal can be dominated by incorrect assignment of contaminant signals to proteins. By contrast, the number of proteins or protein groups identified distinguished most FM1-43$_{high}$ or FM1-43$_{low}$ samples from noise (*Figure 1I*); the samples with low numbers of identifications could represent sorting events where cells missed their target nanowell.

To examine the composition of FM1-43$_{high}$ and FM1-43$_{low}$ samples, we used relative iBAQ (riBAQ) for quantitation (*Shin et al., 2013*; *Krey et al., 2014*) and displayed the 60 most abundant proteins of the 20 cell samples (*Figure 2A*). All identified proteins from the 20 cell samples are displayed in *Figure 2—figure supplement 1*, and all identified proteins from the single-cell samples are displayed in *Figure 2—figure supplements 2* and *3*. We used a volcano plot analysis to show those proteins that had statistically significant enrichment in the 20 cell FM1-43$_{high}$ and FM1-43$_{low}$ samples (*Figure 2B*). Well-known hair-cell proteins were enriched significantly in FM1-43$_{high}$ cells, including the mobile Ca$^{2+}$ buffers OCM and CALB2, as well as the molecular motor MYO6. Several proteins were highly enriched in the FM1-43$_{low}$ samples, including TMSB4X, STMN2, SH3BGRL, and MARCKS.

## Characterization of TMSB4X and monomeric actin in chick utricle

The proteomics experiments also revealed several abundant proteins that had not been previously found to be hair-cell specific, including GSTO1, GPX2, CRABP1, and AK1; TMSB4X and AGR3 were examples of proteins that were much more abundant in supporting cells (*Figure 1—source data 1*; *Figure 2A–B*). We examined several of these proteins in E15 chick utricles using immunocytochemistry. Antibody labeling for AGR3 and the hair-cell marker OTOF labeling did not overlap, and the elongated cell bodies labeled for AGR3 indicated that it marked supporting cells (*Figure 3A–C* and *Figure 3—figure supplement 1*). By contrast, CRABP1 was specific for hair cells, seen by the overlap with OTOF (*Figure 3D–F* and *Figure 3—figure supplement 2*).

The thymosin-beta family of proteins, which includes TMSB4X, are actin monomer binding proteins that sequester substantial fractions of actin in many cell types (*Nachmias, 1993*; *Sun et al., 1995*). Five TMSB4X peptides were identified by mass spectrometry, which covered 75% of the ~5

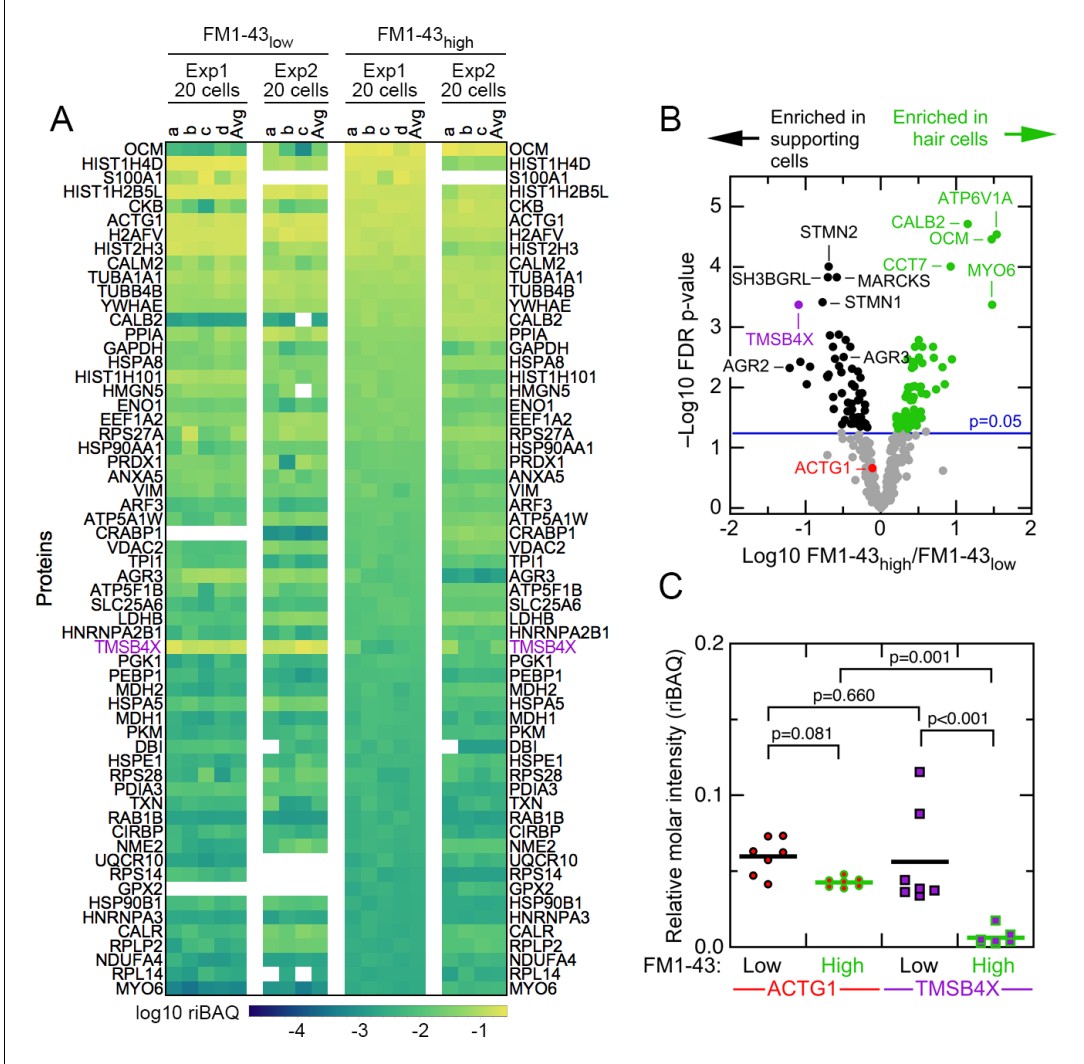

**Figure 2.** Abundant proteins in small pools of isolated E15 chick utricle cells. (**A**) Heat map showing top 60 proteins or protein groups in samples of 20 cells, sorted by the average of the 20 cell FM1-43$_{high}$ samples. FM1-43$_{low}$ and FM1-43$_{high}$ samples from Experiment 1 (Exp1) and Experiment 2 (Exp2) are both displayed, as are the averages of the individual samples. TMSB4X is called out with magenta type. Scale on bottom indicates relationship between riBAQ and color. (**B**) Volcano plot showing relationship between FM1-43$_{high}$/FM1-43$_{low}$ enrichment (x-axis) and false discovery rate (FDR)-adjusted p-value (y-axis). Proteins that are significantly enriched are labeled with green (FM1-43$_{high}$>FM1-43$_{low}$) or black (FM1-43$_{low}$>FM1-43$_{high}$). (**C**) ACTG1 and TMBS4X quantitation. Relative molar fraction (riBAQ) quantitation of ACTG1 (circle, red fill) and TMSB4X (square, magenta fill) expression in FM1-43$_{high}$ cells (green outline) or FM1-43$_{low}$ cells (black outline). Samples with 20 cells are plotted; lines indicate mean expression level for the group. Statistical significance is indicated.
DOI: https://doi.org/10.7554/eLife.50777.006

The following figure supplements are available for figure 2:

**Figure supplement 1.** Expression levels for all proteins in samples from Experiments 1 and 2 that contain pools of three cells, five cells, or 20 cells.
DOI: https://doi.org/10.7554/eLife.50777.007

**Figure supplement 2.** Expression levels for all proteins in samples from Experiments 1 and 2 from single-cell samples.
DOI: https://doi.org/10.7554/eLife.50777.008

**Figure supplement 3.** Expression levels for all proteins in single-cell samples in Experiments 1 and 2.
DOI: https://doi.org/10.7554/eLife.50777.009

kD protein; one of the peptides was shared by TMSB15B, another member of the family (*Figure 1—figure supplement 2*). Analysis of transcript expression in mouse inner ear using gEAR (https://gear.igs.umaryland.edu) indicated that *Tmsb4x* expression was considerably higher than that of another paralog, *Tmsb10*, and much higher than the two *Tmsb15* isoforms, justifying our focus on TMSB4X.

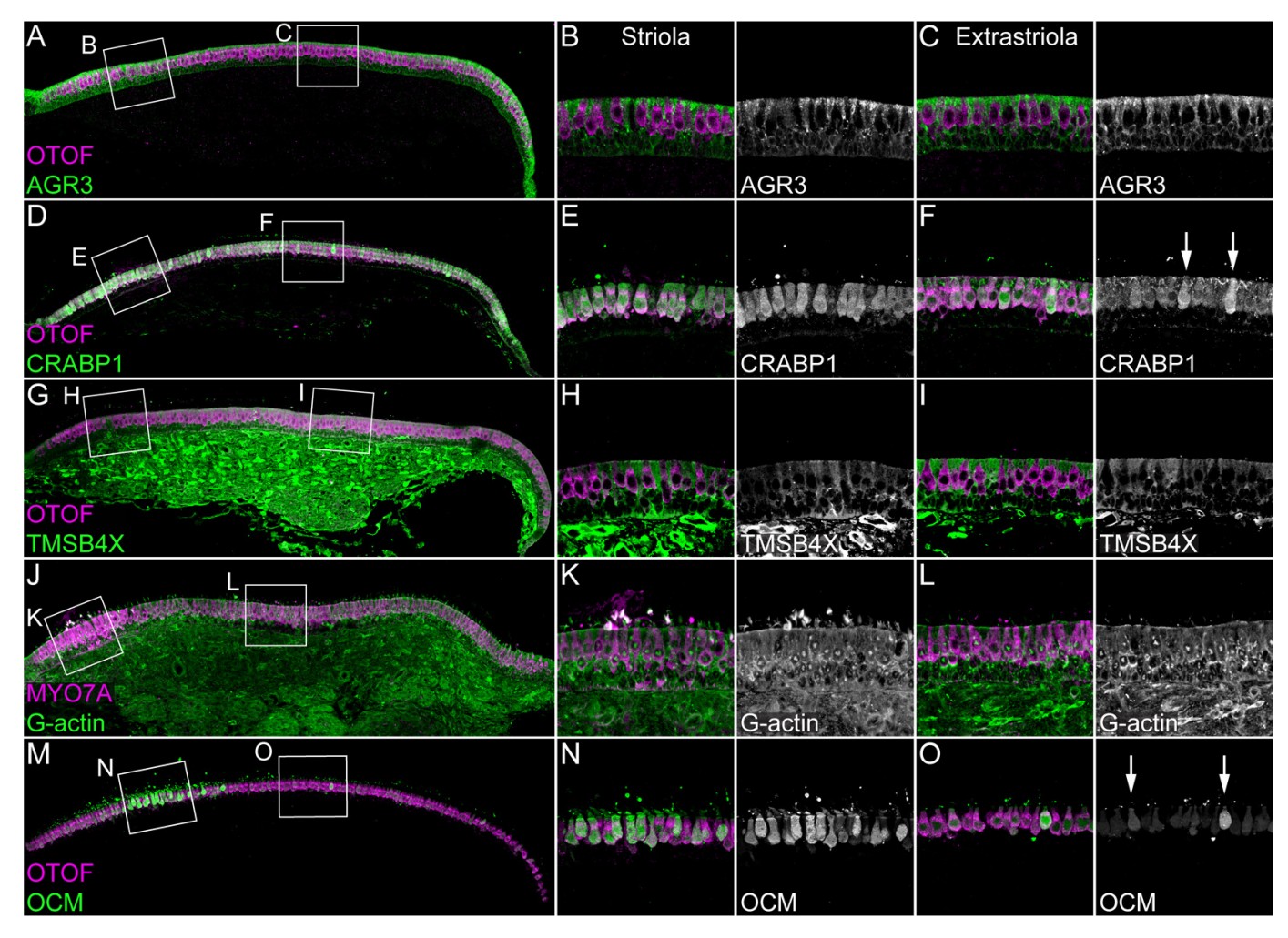

**Figure 3.** Immunolocalization of proteins enriched in hair cells or supporting cells of E15 chick utricle. Confocal z-stacks of vibratome cross-sections of the whole utricle were imaged with the tiling and stitching function in Zeiss ZEN. Confocal z-stacks for magnified extrastriolar and striolar regions were collected separately. A subset of the z-stacks series was used for the maximum intensity projection to preserve single-cell resolution. Hair cells are labeled with antibodies against OTOF or MYO7A (each in magenta). (**A–C**) AGR3 (green) was detected in extrastriolar and striolar supporting cells. Panel full widths: A, 913 µm; B-C, 125 µm. (**D–F**) CRABP1 (green) is concentrated in hair cells. A few extrastriolar hair cells show very high levels of CRABP1 (arrows). Panel full widths: D, 1038 µm; E-F, 125 µm. (**G–I**) TMSB4X (green) immunoreactivity was intense in cells of the mesenchymal stromal cell layer. TMSB4X was detectable at moderate levels in extrastriolar and striolar supporting cells and at low levels in hair cells located in extrastriolar and striolar regions. Panel full widths: G, 934 µm; H-I, 125 µm. (**J–L**) G-actin (JLA20 antibody, green) is expressed at equal levels in hair cells, supporting cells and mesenchymal stromal cells. Panel full widths: J, 839 µm; K-L, 125 µm. (**M–O**) OCM was detectable at high levels in striolar hair cells and at low levels in extrastriolar hair cells. A few extrastriolar hair cells display high levels of OCM (arrows). Panel full widths: M, 946 µm; N-O, 125 µm. Expanded images of all individual channels (transmitted light, nuclei, F-actin, hair cells, and specific antibody) are shown in *Figure 3—figure supplements 1–5*.
DOI: https://doi.org/10.7554/eLife.50777.010

The following figure supplements are available for figure 3:

**Figure supplement 1.** Immunolocalization of AGR3 in E15 chick utricle.
DOI: https://doi.org/10.7554/eLife.50777.011
**Figure supplement 2.** Immunolocalization of CRABP1 in E15 chick utricle.
DOI: https://doi.org/10.7554/eLife.50777.012
**Figure supplement 3.** TMSB4X localization.
DOI: https://doi.org/10.7554/eLife.50777.013
**Figure supplement 4.** G-actin localization with JLA20 antibody.
DOI: https://doi.org/10.7554/eLife.50777.014
**Figure supplement 5.** OCM localization with anti-PV3 antibody.
*Figure 3 continued on next page*

*Figure 3 continued*

DOI: https://doi.org/10.7554/eLife.50777.015

**Figure supplement 6.** Identification of striola and extrastriola regions.

DOI: https://doi.org/10.7554/eLife.50777.016

To localize TMSB4X in the E15 chick utricle, we used an antibody that has been validated previously with knock-down experiments against mouse TMSB4X (*Zhou et al., 2013*; *Li et al., 2018*); chicken TMSB4X differs from mouse and human TMSB4X by only two serine-to-threonine substitutions out of 44 total amino acids. TMSB4X immunoreactivity was cytoplasmic and strong in supporting cells and substantially reduced in hair cells (*Figure 3G–I* and *Figure 3—figure supplement 3*), which was consistent with the mass-spectrometry results. Because TMSB4X maintains actin in a monomeric form (G-actin), probes for G-actin like the JLA20 antibody (*Lin, 1981*) provide another way of localizing the pool of unpolymerized actin. JLA20 immunoreactivity was comparable in most cells, although there was an increased level of signal at the base of the hair cells (*Figure 3J–L* and *Figure 3—figure supplement 4*).

The concentration of TMSB4X relative to total actin should indicate how much free actin is available for assembling filamentous structures like stereocilia (*Weber et al., 1992*). Analyzing the 20 cell samples, we found that the ACTG1 protein group—total actin—accounted for a relative molar fraction (riBAQ) of $0.043 \pm 0.001$ (mean ± SEM) in FM1-43$_{high}$ cells and $0.060 \pm 0.005$ in FM1-43$_{low}$ cells (*Figure 2C*). A mixed-effects model accounting for intra-sample correlations indicated that these concentrations differed significantly, albeit only at an alpha level of 0.05 (summary statistics with confidence intervals are reported in *Table 1*). While TMSB4X accounted for a relative molar fraction of only $0.006 \pm 0.002$ in FM1-43$_{high}$ cells, it was $0.056 \pm 0.012$ in FM1-43$_{low}$ cells, ten-fold higher (*Figure 2C*) and significantly different (p<0.001). Critically, the concentration of hair-cell TMSB4X differed significantly from that of hair-cell actin (p=0.001), while the concentration of supporting cell TMSB4X did not differ from that of supporting cell actin (p=0.660). Because TMSBX and actin interact with a 1:1 stoichiometry (*Goldschmidt-Clermont et al., 1992*), and no other actin-binding proteins are detected at similar high levels, our quantitation suggests that there is enough TMSB4X to bind most actin monomers in supporting cells.

In wholemount preparations, we counted $72 \pm 8$ stereocilia per utricle hair cell (mean ± SD; N = 26 from striolar and extrastriolar regions). The actin quantitation suggested that each E15 hair cell contains ~15,000,000 actin molecules (G- and F-actin combined). If nearly all actin is in stereocilia (*Tilney and Tilney, 1988*), then each stereocilium would contain ~200,000 actin molecules. While fewer than the 400,000 molecules estimated per E20 chick stereocilium (*Shin et al., 2013*), the value is consistent with the relative immaturity of E15 cells.

## Developmental trajectory analysis using single-cell proteomics

While our results showed that the expression profile of pooled-cell samples distinguished between supporting cells and hair cells, a single contaminating cell in a pool could distort the pool's expression pattern. We therefore examined whether we could achieve similar discrimination based on the 30 single-cell profiles, despite the low numbers of identifications in each cell. We used 75 proteins or protein groups that were detected in at least five cells, and normalized and batch-corrected

**Table 1.** Summary statistics with confidence intervals for *Figure 2C*.

| Comparison | Estimate | Std. error | df | t-value | 95% Confidence interval Lower bound | Upper bound | p-value |
|---|---|---|---|---|---|---|---|
| ACTG1 FM-high – ACTG1 FM-low | −0.017 | 0.009 | 22.3 | −1.830 | −0.036 | 0.002 | 0.081 |
| ACTG1 FM-high – TMSB4X FM-high | 0.036 | 0.008 | 12.0 | 4.607 | 0.019 | 0.054 | 0.001 |
| ACTG1 FM-low - TMSB4X FM-low | 0.004 | 0.008 | 12.0 | 0.451 | −0.014 | 0.021 | 0.66 |
| TMSB4X FM-low - TMSB4X FM-low | −0.050 | 0.009 | 22.3 | −5.356 | −0.069 | −0.031 | <0.001 |

DOI: https://doi.org/10.7554/eLife.50777.017

(*Johnson et al., 2007*; *Büttner et al., 2019*; *Luecken and Theis, 2019*) data from individual cells. The resulting expression values, referred to as log2 normalized iBAQ (niBAQ) units, comprised an expression matrix having similar characteristics to single-cell transcriptomics data. Non-detects accounted for 62% of the values, while for 83% of identifications the detected values followed a log-normal distribution (Shapiro-Wilk normality test p>0.05).

Because methods developed for single-cell transcript analysis (*Luecken and Theis, 2019*) should be suitable for dissection of the single-cell proteomics results, we applied CellTrails, which we previously used to uncover the branching trajectory from progenitors to hair cells in the chicken utricle using transcript data (*Ellwanger et al., 2018*). To interpret the latent structure in the single-cell mass spectrometry data, its lower-dimensional manifold was investigated using CellTrails' robust nonlinear spectral embedding on the submatrix of the 37 highest variable identifications (*Figure 4A*). Appropriately, most cells segregated according to their FM1-43 uptake (*Figure 4B*). We noted that the protein pattern of three cells classified as FM1-43$_{high}$ appeared to match better to the FM1-43$_{low}$ (supporting cell) pattern. Similarly, two FM1-43$_{low}$ cells were embedded in the neighborhood of cells with a high FM1-43 uptake (hair cells). While FM1-43 is useful for labeling hair cells, transcript analysis showed that FM1-43 levels are not a perfect proxy for hair-cell maturity (*Ellwanger et al., 2018*); for example, hair cells with damaged mechanotransduction will not load with the dye and such cells would be classified as FM1-43$_{low}$. Alternatively, cells with relatively low FM1-43 could be transitional cells between progenitors and mature hair cells (*Ellwanger et al., 2018*). We therefore surmised that we could elicit a developmental trajectory from the single-cell protein expression patterns. The chronological ordering of the cells was learned in the lower-dimensional manifold and a pseudotime value was assigned to each cell (*Figure 4C*).

The 75 proteins sufficiently detected on single-cell level are all relatively highly expressed and largely do not include those expected to distinguish different classes of hair cells (*Ellwanger et al., 2018*). Moreover, we expect that our sample is dominated by type II hair cells, especially those from extrastriola regions, as they are much more numerous than type I hair cells (*Ellwanger et al., 2018*). Because of the gating strategy used (*Figure 1—figure supplement 1*), we primarily sampled cells from either end of the developmental trajectory, which was apparent from the gap approximately at the midpoint of the pseudotime axis (*Figure 4D*). Nevertheless, we sampled sufficient numbers of differentiating cells to establish a developmental trajectory.

We examined protein expression dynamics as a function of developmental pseudotime (*Figure 4E*). Protein expression changed systematically along the developmental pseudotime axis with the expected trends: proteins enriched in supporting cells, including AGR3 and TMSB4X, decreased in expression along the pseudotime axis (*Figure 4E–F*). By contrast, proteins known to be enriched in hair cells, including OCM, CALB2, MYO6, CKB, and GAPDH, all increased as pseudotime progressed (*Figure 4E–F*).

## Transcriptomic confirmation of TMSB4X enrichment in progenitor cells

We predicted that the decrease in TMSB4X as hair cells mature arose from downregulation of *TMSB4X* transcript expression during differentiation of hair cells. We therefore used transcriptomic profiling of single cells isolated from E15 chick utricle to examine gene expression during the bifurcating trajectory that describes the development of progenitor cells to mature striolar and extrastriolar hair cells (*Ellwanger et al., 2018*). We carried out scRNA-seq transcriptomic profiling using the Smart-seq protocol (*Picelli et al., 2014*) on 384 FACS-sorted E15 chick utricle epithelial cells. To provide maximum correlation of *TMSB4X* expression changes with chicken utricle hair cell maturation, we reconstructed the trajectory in similar fashion as previously described (*Ellwanger et al., 2018*), carrying out the analysis with 182 assay genes already including *GSTO1* and *CRABP1* from that previous study, supplemented with *TMSB4X*, *AGR3*, *GPX2* and *AK1*. Nine cellular subgroups emerged, each of which was distinguished by distinct marker gene sets (*Figure 5A*). Based on their expression profiles, for example the lack of *TECTA* and especially high levels of *TMSB4X* (*Figure 5A*; see also *Figure 3*), two subgroups (S8 and S9) appeared to be stromal cells; to focus on the developmental progression of progenitor (supporting) cells to hair cells, we removed S8 and S9 for subsequent analysis.

We mapped the remaining 254 individual cells of subgroups S1-S7 along developmental trajectories, plotting CellTrails maps (*Ellwanger et al., 2018*) to demonstrate the branching nature of the trajectory (*Figure 5B*). Our assay was biased for hair-bundle genes, and at least half of the cells

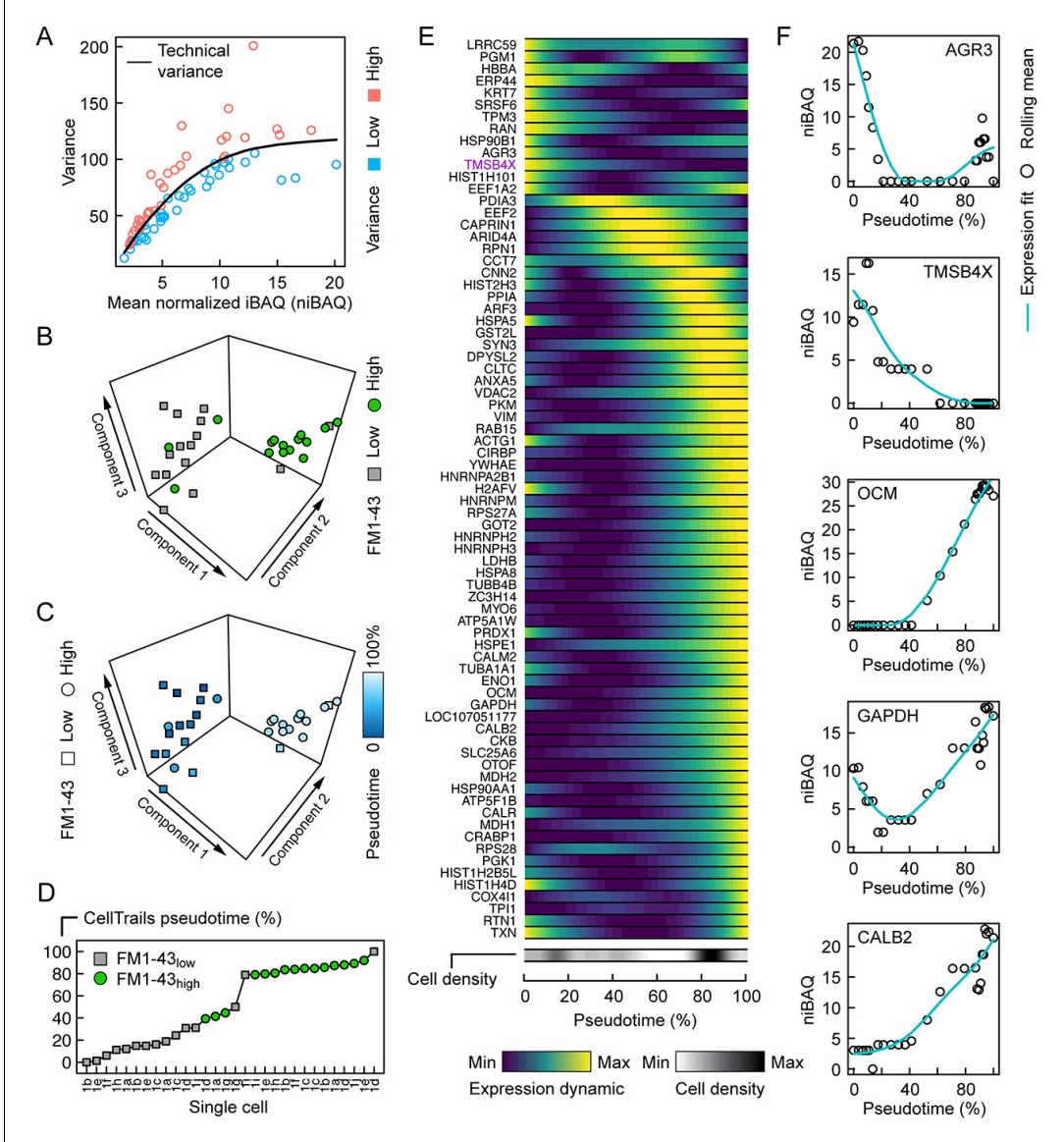

**Figure 4.** Pseudotemporal ordering of single utricle cells based on proteomics measurements. (**A**) Relationship between variance and mean expression, distinguishes proteins or protein groups with low variance (blue) or high variance (salmon). (**B**) First three components of CellTrails' spectral embedding, with FM1-43low (square, gray) and FM1-43high (circle, green) cells indicated. (**C**) First three components of CellTrails' spectral embedding with cells colorized by the inferred pseudotime. (**D**) Chronological ordering of single cells as a function of pseudotime shows that cell ordering correlates with the FM1-43 uptake gradient. (**E**) Scaled expression dynamics over pseudotime for all analyzed proteins or protein groups. A cubic smoothing spline with four degrees of freedom was fit on the rolling mean for each protein. Cell density bar underneath the heat map shows the density of cells along the pseudotime axis. Heat map and cell density scale is shown below. (**F**) Absolute expression dynamics of log2 niBAQ expression levels as a function of pseudotime for various proteins. Blue line is expression fit; circle is the rolling mean for each protein.

DOI: https://doi.org/10.7554/eLife.50777.018

isolated were hair cells, so it is unsurprising that the final trajectory revealed not only the transition from progenitor (supporting) cells to hair cells, but also further developmental branching. One major branch was supporting cells, as these cells expressed markers like *TECTA* and *OTOA* (*Figure 5H*; *Figure 5—source data 1*). The other two major branches were hair cells, as they occurred after the peak of *ATOH1* expression and also showed high levels of hair-cell markers like *MYO7A* (*Figure 5C*; *Figure 5—source data 1*).

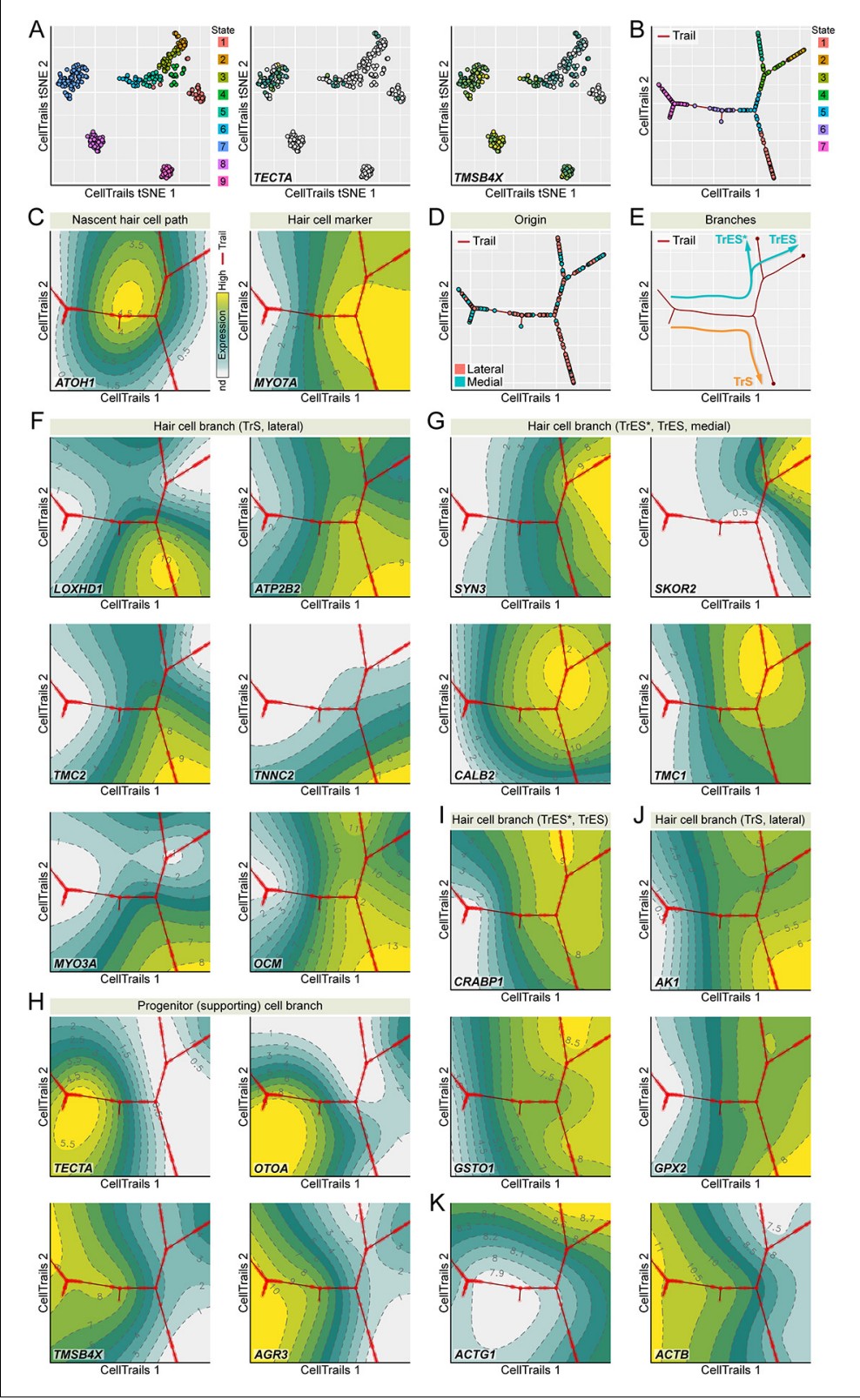

**Figure 5.** Developmental trajectory identified from single utricle cell RNA-seq measurements. (**A**) CellTrails identifies nine distinct states (cellular subgroups). Shown are 328 points representing single utricle cells projected into two-dimensional space using CellTrails t-distributed stochastic neighbor embedding (tSNE). Cells are colored by state affiliation. Two states (S8-9) were classified as stromal cells and excluded from this study based on the

*Figure 5 continued on next page*

*Figure 5 continued*

lack of *TECTA* expression and the high levels of *TMSB4X* expression. (**B**) CellTrails trail map of 254 single chicken utricle cells reveals a bifurcating trajectory. (**C**) *ATOH1* peaks before the main bifurcation of the right major branches. CellTrails map shows that *MYO7A*-expressing hair cells are located downstream of the *ATOH1* peak on the right half of the trajectory map. (**D**) Projection of medial and lateral cell origin metadata into the trail map. Cells from the lateral side accumulate along the lower-right trajectory whereas cells from both halves are located along the upper-right trajectory. (**E**) Predicted developmental extrastriolar (TrES*, TrES), and striolar (TrS) hair cell trajectories. (**F**) *LOXHD1, ATP2B2, TMC2, TNNC2, MYO3A,* and *OCM* expression levels are associated with the lower-right lateral striolar (TrS) branch. (**G**) CellTrails maps showing high expression of *SYN3, SKOR2, CALB2,* and *TMC1* along the upper-left medial extrastriolar (TrES*, TrES) branch. (**H**) Expression of supporting cell marker genes *TECTA, OTOA, TMSB4X* and *AGR3* defines the location of the progenitor (supporting) cell population along the left major branch. (**I**) Expression of *CRABP1* and *GSTO* is associated with the extrastriolar trajectory* (TrES*, see also *Figure 3G*, high *CRABP1* expression extrastriolar hair cells). (**J**) *AK1* and *GPX2* are enriched in the striolar trajectory (TrS). (**K**) *ACTB* expression decreases and *ACTG1* expression increases while hair cells develop.

DOI: https://doi.org/10.7554/eLife.50777.019
The following source data is available for figure 5:

**Source data 1.** CellTrails analysis of single-cell RNA-seq data.
DOI: https://doi.org/10.7554/eLife.50777.020

Enrichment of *LOXHD1, ATP2B2, TMC2, TNNC2, MYO3A* and *OCM* indicated that the branch projecting down and to the right was from striolar cells (*Figure 5F*; *Figure 5—source data 1*), which we named TrS ('trail striola') as previously defined (*Ellwanger et al., 2018*). Similarly, enrichment of *SYN3, SKOR2, CALB2,* and *TMC1* indicated the branch projecting up and to the right is from extra-striolar cells, which we named TrES. TrES branched again, and relative enrichment of *ATP2B2* (*Figure 5F*; *Figure 5—source data 1*) and reduced expression of *SKOR2* (*Figure 5G*; *Figure 5—source data 1*) indicated that the left-hand branch was equivalent to the novel hair cell type TrES* found in our previous study (*Ellwanger et al., 2018*).

To confirm the spatial identity of the two major hair cell branches, all experiments were carried out with E15 chicken utricles split apart into lateral halves, which contain striolar and extrastriolar cells, and medial halves, which contain only extrastriolar cells. The lower-right branch was populated nearly entirely by lateral cells, which confirms that it represents striolar hair cells (*Figure 5D–E*). We conclude that the scRNA-seq experiment accurately replicated our previous experiment using a mul-tiplex RT-qPCR approach (*Ellwanger et al., 2018*).

We next examined the genes highlighted in the proteomics experiments, including *TMSB4X, AGR3, GSTO1, GPX2, CRABP1,* and *AK1*. As predicted from the proteomics and localization experi-ments, the CellTrails analysis showed that *TMSB4X* and *AGR3* were specific to progenitor (support-ing) cells (*Figure 5H*), while *GSTO1, GPX2, CRABP1,* and *AK1* were specific to hair cells (*Figure 5I*). The CellTrails maps indicated that *GPX2* and *AK1* were concentrated in striolar hair cells, while *CRABP1* was enriched in extrastriolar cells, particularly TrES* (*Figure 5I*). Cells observed with high levels of CRABP1 in immunocytochemistry experiments could be the TrES* cells (*Figure 3G,I*). *GSTO1* was expressed at similar levels in both hair cell types. Available antibodies against GPX2, AK1, and GSTO1 were insufficiently reliable to check their hair-cell specificity. Examining databases in gEAR, however, we noted that *Gpx2* and *Ak1* are predicted to be substantially enriched in hair cells as compared to non-hair cells in mouse utricle; by contrast, *Gsto1* is expressed at higher levels in mouse utricle non-hair cells than in hair cells.

The scRNA-seq results corroborated the expression dynamics of TMSB4X on transcriptional level. High in progenitor cells, its transcriptional activity decreased substantially during hair cell differentia-tion (*Figure 5H*), and was nearly undetectable in striolar hair cells. Interestingly, *TMSB4X* was expressed at detectable levels, albeit relatively low, in cells along TrES as compared to TrES* (*Figure 5H*).

We also noted a striking decrease in *ACTB* expression as hair cells differentiated; the CellTrails maps suggested that *ACTB* was >10 fold higher in progenitor cells than in hair cells (*Figure 5K*; *Fig-ure 5—source data 1*). *ACTG1* increased modestly in expression, especially in TrES cells, but overall was present at lower levels than *ACTB* (*Figure 5K*; *Figure 5—source data 1*). Similar trends for these actin isoforms were also seen in our previous data (*Ellwanger et al., 2018*). ACTB and ACTG1

differ by only four amino acids, however, and we only detected one of the peptides that distinguish the isoforms (Ac-EEEIAALVIDNGSGMCK from ACTG1) in single mass spectrometry run. We were therefore unable to accurately measure the relative abundance of the two actin isoforms in our protein mass spectrometry experiments.

## Discussion

Global analysis of proteins from single cells has previously been thwarted by nonspecific adsorption of proteins to surfaces and the lack of a scheme to amplify proteins (*Couvillion et al., 2019*). Recent development of nanowell sample processing with nanoPOTS, coupled with extremely sensitive mass spectrometers, permits detection of abundant proteins of even very small cells (<1 picoliter). We exploited this method to characterize the abundant proteins of FACS-sorted supporting cells, the hair cell progenitors, and hair cells from embryonic chick utricles. Remarkably, we were able to use the mass spectrometry data from single utricle cells to reconstruct a developmental trajectory from protein-expression values alone.

In addition, we identified several proteins not previously highlighted as specific for hair cells (CRABP1, GSTO1, GPX2, AK1) or for supporting cells (AGR3, TMSB4X). TMSB4X was present at nearly equimolar levels with respect to actin in supporting cells and thus may sequester actin monomers there. By contrast, in hair cells, TMSB4X was only one-tenth as abundant as actin. This developmental change was characterized in more depth using single-cell RNA sequencing, which showed that the drop in *TMSB4X* was greater in extrastriolar hair cells than in striolar hair cells. Together, these data are consistent with the hypothesis that downregulation of TMSB4X allows differentiating hair cells to construct their hair bundles with newly available actin monomers.

### Single-cell proteomics detection

While fewer proteins were identified from single hair cells than were identified in a recent report using the same technique with single HeLa cells and primary lung cells (*Zhu et al., 2018a*), hair cells have a volume of ~1 picoliter, while HeLa cells are five times larger in volume. Given this difference, detection of fewer proteins in hair cells than in HeLa cells was not surprising. The 700 proteins detected in HeLa cells are also a substantial underestimate of the total that are present there; using two-dimensional separation prior to mass spectrometry, over 11,000 proteins or protein groups can be detected from these cells (*Geiger et al., 2012*; *Kulak et al., 2017*), and analyses with a one-dimensional separation, like that used here, typically yield ~3000 proteins. The total number of proteins in hair cells and supporting cells is similarly large; using separation in only one chromatographic dimension, we identified ~3000 proteins from E15 chick utricle sensory epithelium (*Shin et al., 2013*). Note that even these numbers are underestimates of the total number of genes expressed; for example, we detected expression from 13,643 genes in our scRNA-seq experiments. The range of protein expression is enormous and the dynamic range of mass spectrometers is limited, which prevents proteomic detection of all expressed genes.

While the nanoPOTS approach is useful for characterizing abundant proteins in small cells or more proteins in larger cells, without further increases in sensitivity, the small number of proteins detected in single hair cells will prevent characterization of low-abundance proteins or deeply categorizing developmental pathways using protein expression. Because the relationship between cell number and total protein signal intensity was nonlinear, especially for 1–3 cell samples (*Figure 1F*), we concluded that we lost significant amounts of protein to adsorption to the nanowells. These results indicate that further improvement of protein recovery is critical to increase proteome coverage and quantification performance of single-cell proteomics technology employing nanoPOTS. Fabricating nanowells with smaller dimensions would be straightforward; however, dispensing single cells by FACS to yet-smaller wells would be difficult to carry out reproducibly. In addition, the nanowell surfaces could be coated chemically with antifouling materials such as polyethylene glycol or poly(2-methyl-2-oxazoline) polymers (*Weydert et al., 2017*). An alternative strategy for single-cell protein analysis uses TMT multiplex labeling with one channel utilized by a sample of several hundred carrier cells, which will reduce the relative error due to protein loss (*Budnik et al., 2018*). Indeed, coupling of TMT multiplex labeling approach with nanoPOTS significantly increases proteome coverage and analysis throughput of single cell proteomics (*Dou et al., 2019*).

We detected much higher levels of small proteins (<20 kD) here than in previous experiments using the same tissue (*Shin et al., 2007*; *Shin et al., 2010*; *Shin et al., 2013*; *Herget et al., 2013*; *Wilmarth et al., 2015*). We attribute this improved detection to the processing of samples in nano-POTS nanowell without transfer steps; other sample preparation methods included steps (e.g., SDS-PAGE gel separation or filter washes) that facilitated loss of small proteins (*Krey et al., 2016*). Our new data indicate that on a molar basis, OCM (*Figure 2A*, *Figure 3M–O*), which was formerly known as parvalbumin 3 in the chick (*Heller et al., 2002*), is the most abundant protein in E15 chick hair cells, accounting for >10% of the total number of protein molecules.

Although we see high levels of small proteins, we do not think that they are over-represented in our data. Moreover, levels of cytoskeletal proteins like actin and tubulin are similar to those seen in bulk proteomics experiments (*Shin et al., 2013*). By contrast, as in most shotgun mass spectrometry experiments, membrane proteins are underrepresented relative to their presence in the genome (*Kar et al., 2017*). Membrane proteins are generally expressed at low levels, however, which also contributes to our inability to see more than a few such proteins (e.g., ATP1A1, SLC17A8, and ATP2B2). Challenges in detecting membrane proteins are not restricted to single-cell proteomics, however.

## Single-cell proteomics trajectory analysis

We report here the first example of the use of single-cell protein mass spectrometry data to construct a developmental trajectory, in this case from progenitors to hair cells. While limited by having only 30 cells for this analysis, CellTrails nevertheless was able to generate a developmental trajectory that discriminated distinct expression patterns of transitioning cells. The proposed trajectory for the single cells studied here accounted for protein expression levels at the endpoints, which represented supporting cells and hair cells; moreover, expression levels of many of the endpoint-selective proteins systematically increased or decreased across the trajectory, as expected. Although we did not detect the branching trajectories of hair cells along the TrS, TrES, and TrES* trails that were seen using multiplex RT-qPCR approach (*Ellwanger et al., 2018*) or scRNA-seq (*Figure 5*), the small number of cells analyzed for protein-based trajectories likely precluded their detection.

Although application of trajectory-analysis methods to single-cell proteomics is very much in its infancy, we show here that CellTrails is suitable for this purpose. The analysis was limited by sensitivity, as proteins detected by mass spectrometry were limited to those expressed at relatively high levels. Improvements of the nanoPOTS method that lead to increased sensitivity and reproducibility will enhance future protein-based trajectory analyses. Nevertheless, while robotic manipulation allows for increased sample-preparation output, the number of cells analyzed presently must remain low because of slow throughput of the mass spectrometry steps. Single-cell RNA-seq approaches are likely to continue to offer much higher throughput and depth for the foreseeable future. That said, analysis of developmental pathways using single-cell proteomics allows the identification of key proteins that change in protein expression level without alternations in transcript levels. Moreover, future single-cell proteomics approaches will allow analysis of posttranslational modifications like phosphorylation, which will expand our ability to probe developmental cascades.

Interestingly, GAPDH was found to increase during hair cell maturation (*Figure 4E–F*), while its mRNA was reported to remain at a constant level (*Avenarius et al., 2014*; *Ellwanger et al., 2018*); this discrepancy suggests that GAPDH undergoes post-transcriptional regulation, either from increased translation efficacy or by protein stabilization following translation. Because GAPDH concentrates in stereocilia (*Shin et al., 2007*; *Shin et al., 2013*), protein stabilization there is the most likely explanation for the increasing GAPDH levels during hair-cell differentiation. This observation highlights the power of the protein-based trajectory analysis; because of the poor correlation between transcript levels and protein levels (*Liu et al., 2016*), understanding how proteins change during a developmental process will require their direct measurement, not inferring their levels as is done with scRNA-seq experiments.

## Actin expression dynamics

During formation of hair cells, we observed a substantial downregulation of *ACTB* during differentiation to hair cells, with the largest decrease in extrastriolar cells; this decrease was partially compensated by an increase in *ACTG1*, especially in extrastriolar cells. If scRNA-seq provides accurate

relative measurements of transcript levels, the sum of *ACTB* and *ACTG1* transcripts in E15 hair cells was one-quarter that in supporting cells. Total actin protein levels were similar in hair cells and supporting cells, however, suggesting that actin synthesis may be reduced during differentiation of hair cells.

In chicken auditory hair cells, ACTB is found nearly exclusively in stereocilia F-actin cores, while the more abundant ACTG1 is distributed to all F-actin assemblies (*Höfer et al., 1997*). Nevertheless, *Actb* or *Actg1* knockout mice have normal stereocilia development, suggesting that the two isoforms are interchangeable in mice, at least initially (*Belyantseva et al., 2009*; *Perrin et al., 2010*). Phenotypes of aging hair cells from the two mutants differ, however, and these differences arise from the ACTB and ACTG1 proteins themselves, not expression dynamics (*Perrin et al., 2010*; *Patrinostro et al., 2018*).

The reduced expression of actin genes is broadly consistent with Tilney's suggestion that each hair cell uses an equivalent-sized bolus of actin to build their hair bundles (*Tilney and Tilney, 1988*). An alternative model dictates that the final amount of actin used in a hair bundle depends on the expression level of crosslinkers like PLS1 and ESPN (*Höfer et al., 1997*; *Sekerková et al., 2011*; *Krey et al., 2016*). A plausible hypothesis incorporating observations reported here is that ACTB is sequestered with TMSB4X in supporting cells; upon differentiation to hair cells, ACTB is made immediately available for stereocilia elongation by degradation of the actin buffer TMSB4X, while ACTG1 expression is increased to provide actin for other assemblies, including the cuticular plate and circumferential actin belt (*Höfer et al., 1997*).

# Materials and methods

**Key resources table**

| Reagent type (species) or resource | Designation | Source or reference | Identifiers | Additional information |
|---|---|---|---|---|
| Biological sample (*Gallus gallus*) | Embryonic day 15 utricle | eggs from Texas A and M University Poultry Science Department | n/a | Freshly isolated from *Gallus gallus* |
| Chemical compound, drug | FM1-43FX | Thermo Fisher | Cat# F35355 | 10 µM |
| Chemical compound, drug | SYTOX Red Dead Cell Stain | Thermo Fisher | Cat# S34859 | 1:1000 (final 5 nM) |
| Chemical compound, drug | thermolysin from geobacillus stearo thermophilus | Sigma-Aldrich | Cat# T7902 | 0.5 mg/ml |
| Chemical compound, drug | Accutase | Innovative Cell Technologies | Cat# AT104 | full strength |
| Chemical compound, drug | n-dodecyl β-D-maltoside | Sigma-Aldrich | Cat# D4641 | 0.1% (w/v) |
| Chemical compound, drug | trypsin | Promega | Cat# V5280 | 10 ng/µl |
| Chemical compound, drug | Lys-C | Promega | Cat# V1671 | 10 ng/µl |
| Chemical compound, drug | DAPI | Thermo Fisher | Cat# D1306 | 1 µg/ml |
| Chemical compound, drug | Alexa Fluor 488-conjugated phalloidin | Thermo Fisher | Cat# A12379 | 1:1000 |
| Antibody | rabbit polyclonal anti-TMSB4X | Proteintech | Cat# 19850–1-AP, RRID: AB_10642437 | 1:250 |
| Antibody | rabbit polyclonal anti-AGR3 | Proteintech | Cat# 11967–1-AP, RRID: n/a | 1:250 |

*Continued on next page*

*Continued*

| Reagent type (species) or resource | Designation | Source or reference | Identifiers | Additional information |
|---|---|---|---|---|
| Antibody | rabbit polyclonal anti-CRABP1 | Proteintech | Cat# 12588–1-AP, RRID: AB_2292271 | 1:250 |
| Antibody | mouse monoclonal anti-G-actin | Developmental Studies Hybridoma Bank | Cat# JLA20, RRID: AB_528068 | 1:250 |
| Antibody | mouse monoclonal anti-otoferlin | Developmental Studies Hybridoma Bank | Cat# HCS-1, RRID: AB_10804296 | 1:250 |
| Antibody | goat polyclonal anti-SOX2 | Santa Cruz Biotechnology | Cat# sc-17320, RRID: AB_2286684 | 1:100 |
| Antibody | mouse monoclonal anti-tubulin beta-3 | BioLegend | Cat# TUJ1, RRID: AB_2313773 | 1:250 |
| Antibody | rabbit polyclonal anti-parvalbumin-3/oncomodulin | Heller laboratory | n/a | 1:1000 |
| Antibody | rabbit polyclonal anti-MYO7A | Proteus Biosciences | Cat# 25–6790, RRID: AB_2314838 | 1:1000 |
| Antibody | Alexa Fluor 546 donkey anti-rabbit polyclonal | Thermo Fisher | Cat# A10040, RRID: AB_2534016 | 1:250 |
| Antibody | Alexa Fluor 647 donkey anti-mouse polyclonal | Thermo Fisher | Cat# A31571, RRID: AB_162542 | 1:100 |
| Antibody | Alexa Fluor 488 donkey anti-goat polyclonal | Thermo Fisher | Cat# A11055, RRID: AB_2534102 | 1:250 |
| Commercial assay or kit | SMARTscribe | Clontech | Cat# 639538 | |
| Commercial assay or kit | Hifi HotStart ReadyMix (2X) | Kapa Biosystems | Cat# KK2602 | |
| Software, algorithm | MaxQuant | Cox lab, Max Planck Institute of Biochemistry | https://www.maxquant.org | |
| Software, algorithm | FIJI (ImageJ) | n/a | http://fiji.sc/ | |
| Software, algorithm | R | The R Project for Statistical Computing | https://www.r-project.org/ | |
| Software, algorithm | limma R package | Bioconductor | DOI: 10.18129/B9.bioc.limma | |
| Software, algorithm | sva R package | Bioconductor | DOI: 10.18129/B9.bioc.sva | |
| Software, algorithm | CellTrails R package | Bioconductor | DOI: 10.18129/B9.bioc.CellTrails | |
| Other | Medium 199 | Thermo Fisher | Cat# 12350039 | |
| Other | SPHERO Drop Delay Calibration Particles | Spherotech | Cat# DDCP-70–2 | |
| Other | nanoPOTS chips | Custom built | n/a | |

## Single-cell collection in Nanowells and sample preparation for proteomics

Single cells were collected from utricles of E15 chick embryos using methods previously described (*Ellwanger et al., 2018*). Utricles were incubated with 10 µM FM1-43FX (Thermo Fisher, Waltham,

MA) for 20 s, then treated with thermolysin (Sigma-Aldrich, St. Louis, MO) to remove the sensory epithelium; cells in the epithelium were dissociated by treating with Accutase (Innovative Cell Technologies, San Diego, CA) with mechanical trituration (*Ellwanger et al., 2018*). Cells were sorted with a BD Influx Instrument (BD Biosciences) set to ''single cell'' mode and equipped with a 85 µm nozzle. To enable the direct cell sorting into nanowells, a customized template was built with the BD FACS software environment (Sortware) to match the format of nanowell array. SPHERO Drop Delay Calibration Particles (Spherotech, Lake Forest, IL) were used to confirm the drop targeting, as well as the parameter optimization. Prior to sorting onto the final collection chip, a separate preparation of E15 chick cells was used to optimize drop delay settings in order to verify alignment of droplets within the well and cell numbers within each well by visual inspection under an epifluorescence microscope. Debris was removed based on forward scatter (FSC) versus side scatter (SSC) and doublets were excluded based on forward scatter (FSC) versus trigger pulse width. SYTOX Red Dead Cell Stain (Thermo Fisher) was added to cells prior to sorting and was used to identify live and dead cells. Based on our final gating approach, which compared 638–1 (SYTOX Red) versus 488–1 (FM1-43), single SYTOX Red-negative FM1-43$_{high}$ or FM1-43$_{low}$ cells were deposited into individual nanowells.

NanoPOTS chips were fabricated on standard microscopy slides as described (*Zhu et al., 2018b*; *Zhu et al., 2018a*). An arrangement of 5 × 13 hydrophilic nanowells was created, each 1 mm diameter with 2.25 mm on-center spacing, and the surrounding chip surface was treated with 2% (v/v) heptadecafluoro-1,1,2,2-tetrahydrodecyl)-dimethylchlorosilane (PFDS) in 2,2,4-trimethylpentane to render it hydrophobic. A glass spacer and cover plate were fabricated for each nanowell chip, allowing the chip to be sealed so that evaporation was minimized during sample incubation. A home-built robotic liquid handling system, capable of subnanoliter dispensing, was used to dispense sample preparation reagents into nanowells (*Zhu et al., 2018b*; *Zhu et al., 2018a*). To lyse cells, and to extract and reduce proteins, 100 nl of 0.2% dodecyl β-D-maltoside containing 5 mM DTT in 0.5x PBS and 25 mM ammonium bicarbonate were added to a nanowell containing FACS-sorted single cells or pools of cells; the chip was then incubated at 70°C for 1 hr. Next, proteins were alkylated using 50 nl of 30 mM iodoacetamide in 50 mM ammonium bicarbonate in the dark at 37°C for 30 min. Lys-C and trypsin (each from Promega, Madison, WI, USA) were added sequentially using 50 nl of 5 ng/µl enzyme solutions in 50 mM ammonium bicarbonate for 4 hr and 6 hr, respectively. Finally, the peptide sample was acidified with 50 nl of 5% formic acid and then collected into a fused-silica capillary (200 µm i.d., 5 cm long). To maximize sample recovery, each nanowell was re-extracted twice, each with 200 nl of 0.1% formic acid in water. The sample collection capillaries were sealed on both ends with Parafilm and stored at −70°C until use.

Two single-cell proteomics experiments were carried out. For Experiment 1, the FM1-43$_{high}$ samples included five samples with 1 cell, four samples with 3 cells, four samples with 5 cells, and four samples with 20 cells; the FM1-43$_{low}$ samples included five samples with 1 cell, three samples with 3 cells, four samples with 5 cells, and four samples with 20 cells. For Experiment 2, the FM1-43$_{high}$ samples included ten samples with 1 cell and four samples with 20 cells; the FM1-43$_{low}$ samples also included ten samples with 1 cell and four samples with 20 cells.

Images of individual FM1-43$_{high}$ and FM1-43$_{low}$ cells (*Figure 1*) were acquired by sorting 1000 cells of either cell type into a drop of 4% paraformaldehyde, which had been placed on a Superfrost Plus glass slide (Fisher Scientific). We used FM1-43 to label the isolated cells after fixation; here, the dye's propensity to insert into the extracellular leaflet of the membrane was used rather than its ability to enter transduction channels. Cells were fixed for 10 min, then a PBS solution containing DAPI (4,6-diamidino-2-phenylindole, 1 µg/ml, Thermo Fisher, D1306) and 10 µM FM1-43FX was added to the slide for 10 min. Fixed and labeled cells were washed, then covered with Vectashield mounting medium (Vector Labs) and imaged using a 63x lens on a Zeiss Elyra PS.1 microscope.

## Other quantitation methods

To quantify cell volume, cells were FACS-sorted (for hair cells and supporting cells), fixed, stained with DAPI and phalloidin, and imaged as described above. For each slice of the z-stack, the Threshold and Make Binary tools of Fiji/ImageJ were used to generate a binary stack, which defined the cell perimeter. The Analyze Particles tool was then used to determine the cell area in each slice. The volume for a slice was calculated as the product of the single-slice area multiplied by the z-stack interval; all slice volumes were added together to estimate total cell volume.

To count stereocilia per hair bundle, Airyscan z-stack images of E15 chicken utricles stained with Alexa Fluor 488-conjugated phalloidin (1:1000, Invitrogen, Carlsbad, CA, A12379) were obtained using a Zeiss LSM 880 microscope; images were acquired near the base of bundles to ensure that all stereocilia were in each image. Stereocilia were manually counted from single x-y images.

## Data-dependent acquisition mass spectrometry

A capillary solid phase extraction (SPE) column (75 µm i.d., with 3 µm C18 particles of 300 Å pore size; Phenomenex, Torrance, USA) was used for initial sample loading and desalting, and was then connected to a 50 cm, 30 µm i.d. column packed with the same material. Mobile phase was delivered at 60 nl/min with a Dionex UltiMate NCP-3200RS pump system (Thermo Fisher). Peptides were separated with a linear 8–22% Buffer B (0.1% formic acid in acetonitrile) gradient over 60 min, followed by a 10 min increase to 45%. The column was washed with 80% Buffer B for 10 min and then equilibrated with 2% Buffer B for 15 min. An Orbitrap Fusion Lumos Tribrid mass spectrometer (Thermo Fisher) was used for data collection. Peptides were ionized at a spray voltage of 2 kV and ions were collected into an ion transfer capillary set at 150°C. The RF lens was set at 30%. MS1 scans used a 375–1575 mass range, a scan resolution of 120,000, an AGC target of $3 \times 10^6$, and a maximum injection time of 246 ms. Precursor ions were selected for MS/MS sequencing if they had charges of +2 to +7 and intensities > 8,000; precursors were isolated with an m/z window of 2 and fragmented by high energy dissociation (HCD) set at 30%. Repeat sampling was reduced by using an exclusion duration of 40 s and m/z tolerance of ± 10 ppm. MS2 scans were carried out in the Orbitrap with an AGC target of $2 \times 10^5$. The maximum injection time and MS2 scan resolution were set as 502 ms and 120,000, respectively.

Andromeda and MaxQuant (version 1.5.3.30) were employed for database searching and label-free protein quantification (*Cox and Mann, 2008*; *Cox et al., 2011*). All MS2 spectra were searched against the NCBI Genome Reference Consortium Chicken Build 6a (GRCg6a) database (49,673 protein sequences; released 2018-03-27). The default MaxQuant contaminants file was edited as described (*Wilmarth et al., 2015*). Carbamidomethylation was selected as fixed modification, and N-terminal protein acetylation and methionine oxidation were set as variable modifications. Peptides were required to contain >5 amino acids and peptide masses must be <4600 Da; two missed cleavages were allowed for each peptide. Peptides and proteins were each filtered with a false discovery rate (FDR) of 0.01. The Match Between Runs algorithm was used to improve proteome coverage; we used an alignment window of 15 min and a match time window of 0.5 min. iBAQ protein intensities were used for quantification.

## Statistical analysis of proteomics data

For enrichment analysis of FM1-43$_{high}$ (N = 4 of 20 cell samples) and FM1-43$_{low}$ (N = 4 of 20 cell samples) cells, the riBAQ data were transformed into log2 scale; linear models with empirical Bayes statistics were fitted to the transformed data using the limma R package (*Ritchie et al., 2015*) (10. 18129/B9.bioc.limma). For the enrichment analysis, we only used the 345 proteins that were measured in at least two replicates in each group. To correct for multiple tests (*Benjamini and Hochberg, 1995*), the FDR was used to correct two-sided p-values from a moderated t-test (*Ritchie et al., 2015*); an enrichment of >1.5 fold and a FDR-adjusted p-value less than 0.05 was considered significant.

For statistical comparisons of ACTG1 and TMSB4X mass spectrometry results, to account for potential intra-sample correlations, a mixed-effects model with a random intercept for samples was fitted to the data and used t-tests of contrasts to assess differences between groups (*Pinheiro and Bates, 2000*). The lmerTest R package (version 3.1–0) was used for the computation (*Kuznetsova et al., 2017*). A p-value less than 0.05 was considered significant.

## Analysis of single-cell proteomics profiles

For single-cell analysis, we filtered all identifications that were robustly detected in at least five cells, leaving a total of 75 proteins (or protein groups). The iBAQ values for proteins in the 30 single cells were normalized to the median of the cells' mean expression and log$_2$-transformed; nondetected values were kept as zeros to avoid imputation artifacts. The empirical Bayes framework available in the *sva* R package (10.18129/B9.bioc.sva) was used to remove the batch covariate, while accounting

for the FM1-43 levels as covariate of interest (*Johnson et al., 2007*; *Büttner et al., 2019*; *Luecken and Theis, 2019*). The same framework has been used previously to correct for batch effects in protein mass spectrometry data (*Carlyle et al., 2017*). The resulting values are referred to as $\log_2$-normalized iBAQ (niBAQ) units. The relationship between protein expression variance and its average expression was fitted using a log-log cubic smoothing spline with four degrees of freedom; 37 proteins with a higher average expression than a $\log_2$ niBAQ value of 1.0 and a higher variance than the fit (*Figure 4A*) were kept for dimensionality reduction. The lower-dimensional manifold (four dimensions) was computed and a trajectory was inferred using the CellTrails R package (*Ellwanger et al., 2018*) (10.18129/B9.bioc.CellTrails). Protein expression dynamics were calculated by fitting the rolling mean (*Haghverdi et al., 2016*; *Keren-Shaul et al., 2017*) with a window size of five of the log2 niBAQ values and the pseudotime axis using a cubic smoothing spline function with 4-degrees of freedom.

## Immunocytochemistry

E15 chicken utricles were dissected in ice-cold Medium 199 containing Hanks' salts (Thermo Fisher) and were fixed with 4% formaldehyde in PBS at 4°C overnight. Utricles were treated with 0.2 M EDTA in PBS until the otoconia became invisible. For embedding, utricles were equilibrated at 50°C in PBS, followed by incubations in 2.5% and 5% low-melt agarose in PBS for 15 min. Utricles were appropriately positioned while the 5% agarose solution was still hot. Cross-sections from E15 utricles were cut to 80 μm thickness with a vibratome (Leica VT1200). Immunocytochemistry was performed as previously described (*Scheibinger et al., 2018*) using the following primary antibodies: rabbit anti-TMSB4X (1:250, Proteintech, Rosemont, IL, 19850–1-AP); rabbit anti-AGR3 (1:250, Proteintech, 11967–1-AP); rabbit anti-CRABP1 (1:250, Proteintech, 12588–1-AP); mouse anti-G-actin (1:250, Developmental Studies Hybridoma Bank, JLA20); mouse anti-otoferlin (1:250, Developmental Studies Hybridoma Bank, HCS-1); goat anti-SOX2 (1:100, Santa Cruz Biotechnology, Dallas, TX, sc-17320); mouse anti-tubulin beta-3 (TUJ1. 1:250, BioLegend, San Diego, CA, 801202); rabbit anti-parvalbumin3/oncomodulin (*Heller et al., 2002*) (1:1000, Heller laboratory); and rabbit anti-MYO7A (1:1000, Proteus Biosciences, Ramona, CA, 25–6790). Secondary antibodies: Alexa Fluor 546 donkey anti-rabbit (1:250, Thermo Fisher, A10040); Alexa Fluor 647 donkey anti-mouse (1:100, Thermo Fisher, A31571); Alexa Fluor 488 donkey anti-goat (1:250, Thermo Fisher, A11055). Validation of antibodies is reported in *Table 2*.

DAPI was used to visualize nuclei and Alexa Fluor 488-phalloidin to visualize F-actin filaments. Sections were imaged with a Plan-Apochromat 40x/1.3 NA oil DIC UV-IRM27 objective on a Zeiss LSM 880 Airyscan laser scanning confocal microscope and Zen Black software. For whole utricle vibratome cross-sections, confocal z-stacks were imaged with the tiling (6% overlap; mode: bounding grid) and stitching function. The 40x Plan-Apochromat objective was used with 0.9x zoom setting. For the extrastriolar and striolar regions, confocal z-stacks were collected separately with the 40x Plan-Apochromat objective used with 1.7 x zoom setting. Extrastriolar and striolar regions in whole utricle vibratome cross-sections were identified using SOX2, TUJ1 and MYO7A antibody labeling as previously described (*Ellwanger et al., 2018*) (*Figure 3—figure supplement 6*). Maturing or mature striolar type I hair cells express MYO7A but lack SOX2 expression and harbor calyx type terminals. Bouton-innervated extrastriolar Type II hair cells express both, MYO7A and SOX2. All supporting cells express SOX2 and lack MYO7A as well as TUJ1 expression (*Figure 3—figure supplement 6*). Maximum intensity projections were generated by a subset of the z-stacks to preserve single cell resolution.

## Single cell isolation and flow cytometry for RNA-seq

Single cells from E15 chicken utricles were collected and sorted as previously described (*Ellwanger et al., 2018*). In this study, FM1-43 labeling was not performed and single cells were sorted with a BD FACSAria Fusion instrument (BD Biosciences). Two independent batches of 270 cells were deposited into individual wells of 96-well plates, with prefilled wells of 4 μl lysis solution with 1 U/μl of recombinant RNase inhibitor (Clontech #2313B), 0.1% Triton X-100 (Thermo #85111), 2.5 mM dNTP (Thermo Fisher #10297018), 2.5 μM oligo d(T)30 VN (5'-AAGCAGTGGTATCAACG-CAGAGTACT30VN-3', IDT). Plates containing sorted cells were immediately sealed, frozen on dry ice and stored at −80°C.

**Table 2.** Antibodies used.

| Antibody | Dilution | Supplier | Catalog # | Validation |
|---|---|---|---|---|
| rabbit anti-TMSB4X | 1:250 | Proteintech | 19850–1-AP | (*Zhou et al., 2013*; *Li et al., 2018*) |
| rabbit anti-AGR3 1:250 | 1:250 | Proteintech | 11967–1-AP | From manufacturer's website: correct-sized band by protein immunoblot in SKOV-3 cells, MCF-7 cells (chicken protein 88% identical to mouse) |
| rabbit anti-CRABP1 1:250 | 1:250 | Proteintech | 12588–1-AP | From manufacturer's website: correct-sized band by protein immunoblot in human spleen tissue, transfected HEK-293 cells (chicken protein 95% identical to human) |
| mouse anti-G-actin 1:250 | 1:250 | Developmental Studies Hybridoma Bank | JLA20 | (*Lin, 1981*) |
| mouse anti-otoferlin 1:250 | 1:250 | Developmental Studies Hybridoma Bank | HCS-1 | (*Goodyear et al., 2010*) (chicken protein 73% identical to mouse) |
| goat anti-SOX2 1:100 | 1:100 | Santa Cruz Biotechnology | sc-17320 | From manufacturer's website: 'recommended for detection of Sox-2 of mouse, rat, human and avian origin by WB, IP, IF, IHC(P) and ELISA' |
| mouse anti-tubulin beta-3 TUJ1. 1:250 | 1:250 | BioLegend | 801202 | (*Lee et al., 1990*) |
| rabbit anti-parvalbumin3/ oncomodulin | 1:1000 | Heller lab | 16910 | (*Heller et al., 2002*) |
| rabbit anti-MYO7A | 1:1000 | Proteus Biosciences | 25–6790 | (*Morgan et al., 2016*) |

DOI: https://doi.org/10.7554/eLife.50777.021

## Single-cell RNA-seq

Single-cell RNA-seq was performed via the method of Picelli and colleagues (*Picelli et al., 2014*) using SMARTscribe (Clontech #639538) for reverse transcription. Kapa Biosystems Hifi HotStart ReadyMix (2X) (#KK2602) was used for 22 cycles of amplification. Amplified cDNAs were purified by bead cleanup using a Biomek FX automated platform and assessed with a fragment analyzer for quantitation and quality assurance. Barcoded libraries were synthesized using a scaled-down Nextera XT protocol (*Mora-Castilla et al., 2016*) in a total volume of 4 µl. A total of 384 libraries were pooled and paired-end sequenced (2 × 150 bp) on a NextSeq 500/550 High Output flow cell. Raw reads in FASTQ format were aligned to the NCBI Gallus gallus v5.0 (GCA_000002315.3) reference genome using custom scripts on the Sherlock Supercomputer Cluster (Stanford). The FastQC tool (version 0.11.6) was used to run an initial quality control check on the raw sequence data. Sequencing reads were mapped by STAR aligner and the transcriptome BAM files were quantified by RSEM. The results were summarized into counts, fragments per kilobase of transcript per million (FPKM), and transcript per million (TPM) expression matrices.

Scater (10.18129/B9.bioc.scater) was used to perform the quality control of the count expression matrix of 384 cells and 19,153 genes. ERCC spike-in transcripts, 56 information-poor cells and 3422 low level expressed genes were removed from the count matrix before read count normalization using SCnorm (*Bacher et al., 2017*). The CellTrails R package (10.18129/B9.bioc.CellTrails) was then utilized following the strategy described in our previous study (*Ellwanger et al., 2018*). The variable trajFeatureNames was set to the 182 previously used assay genes with the addition of *AGR3*, *AK1*, *GPX2*, and *TMSB4X*, which restricted to the analysis to 186 genes (*Table 3*).

## Replicates

In all cases, samples were biological replicates—none of the biological samples were split to be run separately as multiple technical replicates. *Figure 1*. *B-C*, Confocal imaging of FACS-sorted cells. Experiment was carried out 3 times. *D-G*, Characterization of mass spectrometry results. Number of samples for FM1-43$_{high}$: 1 cell, N = 5; 3 cells, N = 4; 5 cells, N = 4; 20 cells, N = 4. Number of samples for FM1-43$_{low}$: 1 cell, N = 5; 3 cells, N = 3; 5 cells, N = 4; 20 cells, N = 4. *H-I*, Number of samples for FM1-43$_{high}$: 1 cell, N = 10; 20 cells, N = 3. Number of samples for FM1-43$_{low}$: 1 cell, N = 10; 20 cells, N = 3. *Figure 2*. Characterization of FM1-43$_{high}$ and FM1-43$_{low}$ samples. *A-C*, Experiment

**Table 3.** Genes included in scRNA-seq CellTrails analysis.

| ABCA5 | (WHRN) | (NSG2) | PLSCR5 | TMSB4X |
|---|---|---|---|---|
| ACO1 | DFNB59 | LOC423919 | PNPT1 | TNNC2 |
| ACTB | (PJVK) | (SHTN1) | PODXL2 | TOLLIP |
| ACTG1 | DIAPH1 | LOC772075 | POU4F3 | TPM1 |
| ACTN1 | DNM1 | (XIRP2) | PPP1R14D | TPM3 |
| ADGRV1 | DPF3 | LOXHD1 | PRPS1 | TPRN |
| AGR3 | DRGX | MAP1A | PTPRQ | TRIOBP |
| AK1 | EFCAB6 | MAPK10 | PTPRT | TTLL12 |
| AKAP5 | EFR3A | MCOLN3 | PTPRZ1 | TUBA3E |
| ANKRD24 | ELMOD1 | MPRIP | RAB26 | TUBAL3 |
| APPL2 | EML1 | MSN | RDX | TUBB2B |
| ARF1 | EPS8L2 | MSRB3 | RFX8 | TUBB6 |
| ARF4 | ESPN | MYH9 | RPS6KA2 | TWF2 |
| ARHGAP17 | ESPNL | MYO15A | RPS6KA5 | USH1C |
| ARMC4 | EZR | MYO1C | RSPH1 | USH1G |
| ATOH1 | FOXJ1 | MYO1H | RSPH9 | USH2A |
| ATP2B1 | FSCN1 | MYO3A | SCG3 | |
| ATP2B2 | FSCN2 | MYO3B | SERPINB6 | |
| ATP2B4 | GALNT9 | MYO6 | SGCB | |
| ATP6V1B2 | GAPDH | MYO7A | SGCG | |
| ATP6V1E1 | GDI2 | NFATC1 | SGIP1 | |
| ATP8B1 | GNAI1 | NMNAT2 | SH3GLB2 | |
| B3GNTL1 | GNAI2 | NPEPPS | SKOR2 | |
| BAIAP2L2 | GNAI3 | OCM | SLC17A8 | |
| BRSK2 | GNAL | OSBP2 | SLC8A1 | |
| CAB39L | GNAS | OSBPL11 | SLC9A3R2 | |
| CACNA2D2 | GNG4 | OSBPL1A | SMPX | |
| CALB2 | GPSM2 | OTOA | SPAG1 | |
| CAPZA1 | GPX2 | OTOF | SPTAN1 | |
| CAPZA2 | GRXCR1 | PAICS | STARD10 | |
| CAPZB | GRXCR2 | PAK1 | STXBP1 | |
| CCDC50 | GSTO1 | PAK2 | SYN3 | |
| CDH23 | HSF5 | PAK3 | TECTA | |
| CHRNA10 | HYDIN | PCDH15 | TECTB | |
| CHRNA9 | IRX2 | PDCD6IP | TMC1 | |
| CIB2 | KIAA1211L | PDK4 | TMC2 | |
| CKB | (KIAA1211) | PDZD7 | TMC5 | |
| CLIC5 | KIAA1549 | PGM2L1 | TMCC2 | |
| CORO2B | KIF1A | PHF21B | TMEM117 | |
| CRABP1 | KLHDC7A | PI4KA | TMEM255B | |
| CSNK2A1 | LCP1 | PITPNA | TMEM30A | |
| CTH | LHFPL5 | PITPNB | TMIE | |
| CUL1 | LHX3 | PLS1 | TMPRSS3 | |
| DFNB31 | LOC416212 | PLS3 | TMPRSS7 | |
| ABCA5 | (WHRN) | (NSG2) | PLSCR5 | TMSB4X |

Table 3 continued

| ABCA5 | (WHRN) | (NSG2) | PLSCR5 | TMSB4X |
|---|---|---|---|---|
| ACO1 | DFNB59 | LOC423919 | PNPT1 | TNNC2 |
| ACTB | (PJVK) | (SHTN1) | PODXL2 | TOLLIP |
| ACTG1 | DIAPH1 | LOC772075 | POU4F3 | TPM1 |
| ACTN1 | DNM1 | (XIRP2) | PPP1R14D | TPM3 |
| ADGRV1 | DPF3 | LOXHD1 | PRPS1 | TPRN |
| AGR3 | DRGX | MAP1A | PTPRQ | TRIOBP |
| AK1 | EFCAB6 | MAPK10 | PTPRT | TTLL12 |
| AKAP5 | EFR3A | MCOLN3 | PTPRZ1 | TUBA3E |
| ANKRD24 | ELMOD1 | MPRIP | RAB26 | TUBAL3 |
| APPL2 | EML1 | MSN | RDX | TUBB2B |
| ARF1 | EPS8L2 | MSRB3 | RFX8 | TUBB6 |
| ARF4 | ESPN | MYH9 | RPS6KA2 | TWF2 |
| ARHGAP17 | ESPNL | MYO15A | RPS6KA5 | USH1C |
| ARMC4 | EZR | MYO1C | RSPH1 | USH1G |
| ATOH1 | FOXJ1 | MYO1H | RSPH9 | USH2A |
| ATP2B1 | FSCN1 | MYO3A | SCG3 | |
| ATP2B2 | FSCN2 | MYO3B | SERPINB6 | |
| ATP2B4 | GALNT9 | MYO6 | SGCB | |
| ATP6V1B2 | GAPDH | MYO7A | SGCG | |
| ATP6V1E1 | GDI2 | NFATC1 | SGIP1 | |
| ATP8B1 | GNAI1 | NMNAT2 | SH3GLB2 | |
| B3GNTL1 | GNAI2 | NPEPPS | SKOR2 | |
| BAIAP2L2 | GNAI3 | OCM | SLC17A8 | |
| BRSK2 | GNAL | OSBP2 | SLC8A1 | |
| CAB39L | GNAS | OSBPL11 | SLC9A3R2 | |
| CACNA2D2 | GNG4 | OSBPL1A | SMPX | |
| CALB2 | GPSM2 | OTOA | SPAG1 | |
| CAPZA1 | GPX2 | OTOF | SPTAN1 | |
| CAPZA2 | GRXCR1 | PAICS | STARD10 | |
| CAPZB | GRXCR2 | PAK1 | STXBP1 | |
| CCDC50 | GSTO1 | PAK2 | SYN3 | |
| CDH23 | HSF5 | PAK3 | TECTA | |
| CHRNA10 | HYDIN | PCDH15 | TECTB | |
| CHRNA9 | IRX2 | PDCD6IP | TMC1 | |
| CIB2 | KIAA1211L | PDK4 | TMC2 | |
| CKB | (KIAA1211) | PDZD7 | TMC5 | |
| CLIC5 | KIAA1549 | PGM2L1 | TMCC2 | |
| CORO2B | KIF1A | PHF21B | TMEM117 | |
| CRABP1 | KLHDC7A | PI4KA | TMEM255B | |
| CSNK2A1 | LCP1 | PITPNA | TMEM30A | |
| CTH | LHFPL5 | PITPNB | TMIE | |
| CUL1 | LHX3 | PLS1 | TMPRSS3 | |
| DFNB31 | LOC416212 | PLS3 | TMPRSS7 | |

DOI: https://doi.org/10.7554/eLife.50777.022

was carried out two times. Panels were based on Experiments 1 and 2. *Figure 3*. *A-C*, Immunofluorescence detection of AGR3. Experiment was carried out 3 times. *D-F*, CRABP1. Experiment was carried out 3 times. *G-I*, TMSB4X. Experiment was carried out 5 times. *J-L*, G-actin. Experiment was carried out 3 times. *M-O*, OCM. Experiment was carried out 3 times. *Figure 4*. Developmental trajectory based on proteomics of single cells. Analysis used 30 single cells from two experiments. *Figure 5*. Developmental trajectory based on scRNA-seq of single cells. Analysis used two independent batches of 270 cells.

## Data availability

The mass spectrometry proteomics data, including raw data from the mass spectrometry runs, have been deposited to the ProteomeXchange Consortium via the PRIDE partner repository (*Perez-Riverol et al., 2019*) with the dataset identifier PXD014256. The analyzed data are reported in *Figure 1—source data 1*. The analyzed single-cell RNA-seq data are reported in *Figure 5—source data 1*.

## Code availability

CellTrails (10.18129/B9.bioc.CellTrails) software is available from Bioconductor (release 3.9). CellTrails is described in detail elsewhere (*Ellwanger et al., 2018*).

## Acknowledgements

We thank Pamela Canaday and Matthew Lewis for their assistance with flow cytometry. We received support from the following OHSU core facilities: FACS from the OHSU Flow Cytometry Shared Resource (supported in part by the OHSU Knight Cancer Institute and the NCI Cancer Center Support Grant P30 CA069533) and confocal microscopy from the OHSU Advanced Light Microscopy Core (P30 NS061800 provided support for imaging). We also received support from the following Stanford core facilities: the Stanford Shared FACS Facility, the Stanford Functional Genomics Facility, and the Otolaryngology Imaging Core. Some of the computing for this project was performed on the Sherlock cluster; we thank the Stanford Research Computing Center for providing computational resources and support. YZ was supported by Earth and Biological Sciences Directorate Mission Seed under the Laboratory Directed Research and Development Program at PNNL, and the Precision Medicine Innovation Co-Laboratory (PMedIC), a joint research collaboration of OHSU and PNNL. A portion of the research at PNNL was performed using EMSL (grid.436923.9), a DOE Office of Science User Facility sponsored by the Office of Biological and Environmental Research. SH was supported by R01 DC015201, by the Hearing Health Foundation's Hearing Restoration Project, and through the Stanford Initiative to Cure Hearing Loss. RTK was supported by R33 CA225248. PGBG was supported by NIH grant R01 DC011034.

## Additional information

### Competing interests

Daniel Christian Ellwanger: is affiliated with Amgen Inc. The author has no other competing interests to declare. The other authors declare that no competing interests exist.

### Funding

| Funder | Grant reference number | Author |
|---|---|---|
| National Institutes of Health | R01 DC011034 | Peter G Barr-Gillespie |
| National Institutes of Health | R01 DC015201 | Stefan Heller |
| National Institutes of Health | R33 CA225248 | Ryan T Kelly |
| U.S. Department of Energy | Laboratory Directed Research and Development Program (Earth & Biological Sciences Directorate Mission Seed) | Ying Zhu |

| | | |
|---|---|---|
| Hearing Health Foundation | Hearing Restoration Project | Stefan Heller |
| Stanford Initiative to Cure Hearing Loss | | Stefan Heller |

The funders had no role in study design, data collection and interpretation, or the decision to submit the work for publication.

### Author contributions

Ying Zhu, Peter G Barr-Gillespie, Conceptualization, Formal analysis, Supervision, Funding acquisition, Investigation, Visualization, Methodology, Writing—original draft, Project administration, Writing—review and editing; Mirko Scheibinger, Conceptualization, Formal analysis, Supervision, Validation, Investigation, Visualization, Methodology, Writing—review and editing; Daniel Christian Ellwanger, Conceptualization, Formal analysis, Validation, Investigation, Visualization, Methodology, Writing—review and editing; Jocelyn F Krey, Conceptualization, Formal analysis, Investigation, Methodology, Writing—review and editing; Dongseok Choi, Conceptualization, Formal analysis, Supervision, Funding acquisition, Visualization, Methodology, Project administration, Writing— review and editing; Ryan T Kelly, Conceptualization, Supervision, Funding acquisition, Investigation, Visualization, Methodology, Writing—review and editing; Stefan Heller, Conceptualization, Formal analysis, Supervision, Funding acquisition, Validation, Visualization, Methodology, Project administration, Writing—review and editing

### Author ORCIDs

Peter G Barr-Gillespie https://orcid.org/0000-0002-9787-5860

### Decision letter and Author response

Decision letter https://doi.org/10.7554/eLife.50777.028
Author response https://doi.org/10.7554/eLife.50777.029

## Additional files

### Supplementary files

• Reporting standard 1. Reporting guidelines for mass spectrometry.
DOI: https://doi.org/10.7554/eLife.50777.023
• Transparent reporting form DOI: https://doi.org/10.7554/eLife.50777.024

### Data availability

The mass spectrometry proteomics data, including raw data from the mass spectrometry runs, have been deposited to the ProteomeXchange Consortium via the PRIDE partner repository with the dataset identifier PXD014256. The analyzed data are reported in Figure 1—source data 1. The analyzed single-cell RNA-seq data are reported in Figure 5—source data 1.

The following dataset was generated:

| Author(s) | Year | Dataset title | Dataset URL | Database and Identifier |
|---|---|---|---|---|
| Barr-Gillespie PG | 2019 | Embryonic day 15 chick utricle single cell analysis | https://www.ebi.ac.uk/pride/archive/projects/PXD014256 | PRIDE, PXD014256 |

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
