## [Decision Letter]

Thank you for submitting your article "Single-cell proteomics reveals downregulation of TMSB4X to drive actin release for stereocilia assembly" for consideration by *eLife*. Your article has been reviewed by two peer reviewers and the evaluation has been overseen by Anna Akhmanova as the Senior Editor. The reviewers have opted to remain anonymous.

The reviewers have discussed the reviews with one another and the Reviewing Editor has drafted this decision to help you prepare a revised submission.

Summary:

The manuscript "Single-cell proteomics reveals downregulation of TMSB4X to drive actin release for stereocilia assembly" presents a series of state-of-the-art protein spectrometry experiments on inner-ear hair cells that provide single-cell proteomes enabling the identification of important steps in hair-cell differentiation. The authors focus on E15 chicken utricle tissue, that includes both supporting and hair cells. Various sample preparation strategies and analysis protocols were used to identify about 600 proteins or protein groups for FM1-43_high_ samples, and about 300 or so for FM1-43_low_. Several proteins specific to either presumptive hair cells or supporting cells were identified, and further experiments were done for some of these proteins to validate their localization. In addition, TMSB4X, found predominantly in supporting cells but not hair-cells, is suggested to play a key role in capturing actin in supporting cells and then releasing it during differentiation to facilitate the assembly of the characteristic actin-based sterocilia bundles of hair cells. Proteomics data also permitted the classification of cells according to a pseudotemporal ordering, and single cell RNA-seq measurements identified developmental trajectories for extrastriolar and striolar hair cells.

Overall, the reviewers felt that this study had merit and was worthy of consideration for publication in *eLife*. In general, the most significant concern of both reviewers was the conclusion that *TMSB4X* actually regulates actin release related to stereocilia assembly. While this is a very reasonable hypothesis, in the absence of experimental data, it seems premature to include this idea in the title of the study.

In addition, one of the reviewers requested some additional clarification regarding the types of proteins that might not be detected in hair cells and supporting cells.

Essential revisions:

1) While the authors suggest that their results show that *TMSB4X* plays a role in releasing actin that can then be used to build stereocilia bundles, there is no evidence for this beyond changes in *TMSB4X* expression data. In light of this, the study then fundamentally represents the demonstration of a technique, single cell proteomic analysis. The title is a clear overstep based on the data provided. There is no direct evidence that TMSB4X actually drives actin release related to stereocilia assembly. This may in fact be what occurs, but without data, it seems inappropriate to include this in the title.

The enrichment of TMSB4X in supporting cells is intriguing and well documented, yet lack of TMSB4X in hair cells is only correlated with the appearance of a hair bundle, causality is not really demonstrated although it is implied, which might be problematic. Whether TMSB4X really functions in capturing actin in supporting cells is an attractive hypothesis, and it should be referred as such (where is the immunostaining of G-actin and TMSB4X showing overlap? Is the antibody for TMSB4X really reliable? It seems to stain a lot of other stuff. Could it be that the hair bundle forms even in the presence of TMSB4X and that other factors are really governing this transition?). Mechanistic details about how the TMSB4X proteins function and get degraded during development are lacking. One can see how obtaining these mechanistic insights is likely beyond the scope of the paper, but perhaps clarifying that this is only a correlation, and that the TMSB4X function in supporting cells is hypothesized but that it has not really been established, would be helpful.

2) There are about 600 proteins identified for hair-cells, and significantly less for supporting cells. An estimate of how much is missing can be obtained by comparing to results from HeLa cells (700 proteins), and the authors do acknowledge that some proteins are missing. This needs to be discussed in more depth, both in terms of what types of proteins might be missing, and how many. For instance, are membrane/soluble proteins well represented in the list? Is the method biasing the detections of some, but not others? Is the mass spec analysis biasing the detection towards small proteins? (I assume orbitrap limits on mass are bypassed by proteolysis treatments, but this could be discussed a bit more). In addition, an upper boundary on the number of different proteins expressed in a cell could also be obtained from mRNA transcripts identified (with obvious caveats, but a good upper limit nonetheless), which could help provide another estimate of how many proteins are missing in the analyses.

---

## [Author Response]

Essential revisions:1) While the authors suggest that their results show that TMSB4X plays a role in releasing actin that can then be used to build stereocilia bundles, there is no evidence for this beyond changes in TMSB4X expression data. In light of this, the study then fundamentally represents the demonstration of a technique, single cell proteomic analysis. The title is a clear overstep based on the data provided. There is no direct evidence that TMSB4X actually drives actin release related to stereocilia assembly. This may in fact be what occurs, but without data, it seems inappropriate to include this in the title.

We agree with the reviewers and have changed the title.

The enrichment of TMSB4X in supporting cells is intriguing and well documented, yet lack of TMSB4X in hair cells is only correlated with the appearance of a hair bundle, causality is not really demonstrated although it is implied, which might be problematic.

Agreed. We have also downplayed the TMSB4X model throughout (see below).

Whether TMSB4X really functions in capturing actin in supporting cells is an attractive hypothesis, and it should be referred as such (where is the immunostaining of G-actin and TMSB4X showing overlap?

We have referred to this idea as a hypothesis throughout the manuscript.

Is the antibody for TMSB4X really reliable? It seems to stain a lot of other stuff.

TMSB4X is expressed widely. The reviewers may be noting the strong staining in the stroma cells in Figure 3H-I and Figure 3—figure supplement 2. However, we also note that there are two groups of cells in the scRNA-seq experiment (clusters S8 and S9 in Figure 5A) that have high expression of TMSB4X (Figure 5A). These cells were removed for the subsequent analysis of scRNA-seq data, but they at least partially corroborate our suggestion that the antibody is specific. The antibody has been validated in knock-down experiments, as indicated in the text.

Could it be that the hair bundle forms even in the presence of TMSB4X and that other factors are really governing this transition?). Mechanistic details about how the TMSB4X proteins function and get degraded during development are lacking. One can see how obtaining these mechanistic insights is likely beyond the scope of the paper, but perhaps clarifying that this is only a correlation, and that the TMSB4X function in supporting cells is hypothesized but that it has not really been established, would be helpful.

We agree with the reviewers that there are several very interesting questions to address beyond the scope of this paper. We also are convinced by the reviewers to indicate clearly that the role we propose for TMSB4X is hypothetical as well.

Last half of Abstract changed to: "The single-cell proteomes enabled the de novo reconstruction of a developmental trajectory using protein expression levels, revealing proteins that greatly increased in expression during differentiation of hair cells (OCM, CRABP1, GPX2, AK1, GSTO1) and those that decreased during differentiation (TMSB4X, AGR3). […] Single-cell proteomics data thus can be mined to reveal features of cellular development that may be missed with transcriptomics."

Last sentence of Introduction changed to: "These data are consistent with a model that suggests that existing monomeric actin is made available for hair-bundle assembly by the degradation of TMSB4X."

Discussion:"TMSB4X was present at nearly equimolar levels with respect to actin in supporting cells and thus may sequester actin monomers there. […] Together, these data are consistent with the hypothesis that downregulation of TMSB4X allows differentiating hair cells to construct their hair bundles with newly available actin monomers."

Discussion: The section titled "TMSB4X dynamics and assembly of stereocilia" was nearly completely removed.

2) There are about 600 proteins identified for hair-cells, and significantly less for supporting cells. An estimate of how much is missing can be obtained by comparing to results from HeLa cells (700 proteins), and the authors do acknowledge that some proteins are missing. This needs to be discussed in more depth, both in terms of what types of proteins might be missing, and how many.

Most cells are thought to express ~10,000 different proteins, not accounting for splice variants; mass spectrometry only detects this many proteins with substantial protein loads, however. Using 20 µg of starting material and two-dimensional separation, the Mann group identified 11,148 proteins or protein groups from HeLa cells (Geiger et al., 2012; Kulak et al., 2017). With a single separation dimension, ~3000 proteins can usually be detected.

For example, in our experiments carrying out proteomics measurements on whole chick utricle sensory epithelia at E20 (Shin et al., 2013), which used far more material, we detected 2753 proteins in at least two of four replicates. We used ~10 utricles per sample, which represents ~400,000 cells. These preparations will contain both hair cells and supporting cells, but very few other cell types.

Of course, the dynamic range of protein expression is enormous (10^7^- to 10^10^-fold) and so most techniques only a capture a slice of the range. The single cell experiments, because they use so little input protein, are necessarily biased against detecting rare proteins. Nevertheless, we can gain significant insight into the function of a cell just by looking at its abundant proteins, as we show here.

For instance, are membrane/soluble proteins well represented in the list? Is the method biasing the detections of some, but not others?

Despite accounting for ~1/3 of all proteins, as with most mass spectrometry approaches, membrane proteins are not represented well on our list of detected proteins. Their hydrophobic peptides do not perform well on reverse-phase liquid chromatography, are not efficiently ionized before introduction into the mass spectrometer, or both. Soluble and cytoskeletal proteins are well represented, but they are also typically of much higher concentration than are membrane proteins.

Is the mass spec analysis biasing the detection towards small proteins? (I assume orbitrap limits on mass are bypassed by proteolysis treatments, but this could be discussed a bit more).

Most mass spectrometry sample preparation methods are biased against small proteins, as those methods typically include sieving steps like gel or filter-based purification. Thus we suggest that our sample prep method provides a more accurate estimate of the levels of these small proteins.

Because of high intrinsic variability for detection efficiency from peptide to peptide, where only a fraction of peptides (5-10%) perform exceptionally well in the mass spectrometer, small proteins can be missed if none of their tryptic peptides are well behaving. Again, normally mass spectrometry analysis is biased against small proteins, so anything that levels the field – like single-pot sample prep – gives a better representation of small proteins.

In addition, an upper boundary on the number of different proteins expressed in a cell could also be obtained from mRNA transcripts identified (with obvious caveats, but a good upper limit nonetheless), which could help provide another estimate of how many proteins are missing in the analyses.

In the scRNA-seq experiments, we detected expression of 13,643 genes, not accounting for splice variants, again in the combination of hair cells and supporting cells. Note that all of the concerns raised in point #2 are good ones, and so we have added an additional text to the Discussion that discusses the depth of protein sequencing and biases for and against certain classes of proteins.